# ASSESSING OPEN-WORLD FORGETTING IN GENERATIVE IMAGE MODEL CUSTOMIZATION

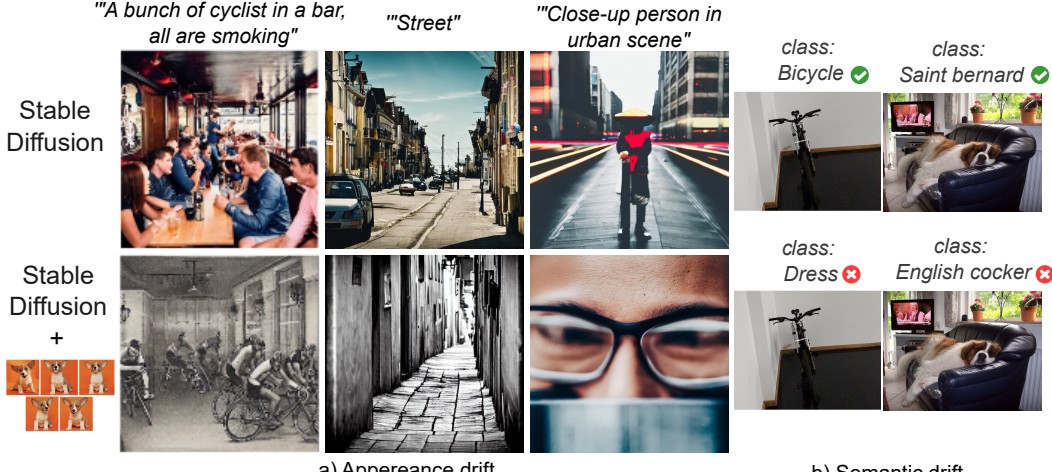

Figure 1: **Unintended consequences in diffusion model customization**. Methods like Dreambooth lead to substantial drift in previously learned representations during the finetuning process even when adapting to as few as five images: a) Appearance drift: Columns demonstrate fine-grained class changes, complete object and scene shifts, and alterations in color (on both rows, images are generated from same seed). b) Semantic drift: finetuning negatively impacts the zero-shot classification capabilities of the models.

## ABSTRACT

Recent advances in diffusion models have significantly enhanced image generation capabilities. However, customizing these models with new classes often leads to unintended consequences that compromise their reliability. We introduce the concept of *open-world forgetting* to emphasize the vast scope of these unintended alterations, contrasting it with the well-studied *closed-world forgetting*, which is measurable by evaluating performance on a limited set of classes or skills. Our research presents the first comprehensive investigation into open-world forgetting in diffusion models, focusing on semantic and appearance drift of representations. We utilize zero-shot classification to analyze semantic drift, revealing that even minor model adaptations lead to unpredictable shifts affecting areas far beyond newly introduced concepts, with dramatic drops in zero-shot classification of up to 60%. Additionally, we observe significant changes in texture and color of generated content when analyzing appearance drift. To address these issues, we propose a mitigation strategy based on functional regularization, designed to preserve original capabilities while accommodating new concepts. Our study aims to raise awareness of unintended changes due to model customization and advocates for the analysis of open-world forgetting in future research on model customization and finetuning methods. Furthermore, we provide insights for developing more robust adaptation methodologies.

# 1 INTRODUCTION

Recent advancements in image generation have led to the development of remarkably powerful foundational models capable of synthesizing highly realistic and diverse visual content. Techniques such as Generative Adversarial Networks (GANs) (Goodfellow et al., 2014), and more recently autoregressive models (Yu et al., 2022), Rectified Flows (Liu et al., 2023), and Denoising Diffusion Probabilistic Models (DDPMs) (Ho et al., 2020), have each contributed to significant progress in the field. These methods offer unique strengths in sample quality, diversity, and controllability. Among them, diffusion models have gained particular prominence due to their recent successes and growing influence, especially in enabling text-based image generation (Shonenkov et al., 2023; Ramesh et al., 2022) and complementary multimodal conditioning (Zhang & Agrawala, 2023; Mou et al., 2023), making them a key focus in current research and applications.

Given the high-quality image generation capabilities of these models, a major focus of research has been on how to efficiently incorporate new content and adapt them to new tasks and domains. To address this challenge, various state-of-the-art transfer learning methods have been introduced. These include finetuning approaches such as DreamBooth (Ruiz et al., 2023b) and CustomDiffusion (Kumari et al., 2023), which allow models to learn new concepts effectively. Conditioning-based methods like ControlNet (Zhang & Agrawala, 2023) and IP-Adapters (Ye et al., 2023) enable precise control over generated images by incorporating additional guidance signals. Prompt methods like Prompt-to-Prompt (Hertz et al., 2023) and Textual Inversion (Gal et al., 2023) enable semantic image editing and learning new concepts without modifying the base model. Parameter-efficient techniques such as Low-Rank Adaptations (LoRA) (Hu et al., 2022) have shown great promise, allowing for rapid adaptation with minimal computational overhead (Blattmann et al., 2023; Shi et al., 2023b). These techniques enable models to learn new concepts effectively, even with few examples.

These methods for adapting diffusion models (Ruiz et al., 2023b; Kumari et al., 2023) mainly rely on transfer learning and primarily focus on finetuning model weights to accommodate newly introduced data. However, they lead to unforeseen changes in the model's behavior, which can have significant implications, altering the model's existing knowledge, and skills, or the alignment between language and visual content within the network. The field of continual learning has long studied the issue of *catastrophic forgetting* in neural networks when these aim to adapt to new data (often referred to as new tasks) (Kirkpatrick et al., 2017; De Lange et al., 2021). Traditionally, this field has focused on what we term *closed-world forgetting*, where evaluation is limited to a fixed set of classes encountered in previously learned tasks or skills. This setting assumes a clear, predefined set of concepts to evaluate against. In contrast, modern foundation models introduce what we term *open-world forgetting*: degradation of the model's capabilities across its vast, unconstrained knowledge space. Unlike closed-world settings, open-world forgetting is particularly challenging to measure since the model's prior knowledge spans countless concepts, making it impossible to exhaustively evaluate what has been forgotten or altered during the adaptation process.

In this paper, we focus on two popular personalization methods, namely Dreambooth (Ruiz et al., 2023b) and CustomDiffusion (Kumari et al., 2023), for a case study of open-world forgetting. These techniques are especially relevant, as they only add very little new knowledge to the network: a single new concept represented by a small set of typically 3-5 images. Although one might expect that finetuning the model with such limited data would have minimal impact on the vast knowledge of the foundation model (e.g., Stable Diffusion), our analysis reveals that even these small updates can lead to highly detrimental consequences. As Figure 1 illustrates, finetuning can drastically alter the image representation of concepts seemingly unrelated to the training images. The complexity of the forgetting underscores the need for a better understanding of how and where it occurs. Without this understanding, finetuned models risk becoming less reliable, less robust, and ultimately less trustworthy, particularly in safety-critical applications where precision and predictability are paramount.

We propose to analyze open-world forgetting from several perspectives. First, we examine *semantic drift* using the recent observation that diffusion models can function as zero-shot classifiers; we propose to compare zero-shot capacity of models before and after adaptation on a set of image classification data sets. Second, we analyze *appearance drift* by evaluating changes in color and perceptual measurements before and after adaptation. Lastly, we assess the extent of forgetting in closely related concepts (*local drift*) versus unrelated concepts. To address these three aspects of drift, we explore a straightforward, yet effective mitigation strategy by introducing a regularization

technique during the training of new concepts. In conclusion, the main contributions of this work are:

- We are the first to systematically analyze *open-world forgetting* in diffusion models due to model adaptation. Results show that even when adapting to very small domains, the consequences can be highly detrimental.

- We propose two approaches to analyze *open-world forgetting*, which are designed to assess *semantic* and *appearance drift* caused by the adaptation. We leverage the zero-shot classification capabilities of diffusion models to measure the semantic drift, and observe drastic performance drops (of over 60% for some classes). Appearance drift analysis confirms that customization leads to considerable changes in intra-class representation, color, and texture.

- We introduce a method to mitigate open-world forgetting, addressing the challenges of observed drift in text-to-image (T2I) models. This method aims to preserve the original model's capabilities while allowing for effective customization. Experiments confirm that it greatly reduces both the semantic and appearance drift caused by open-world forgetting.

## 2  RELATED WORK

**Text-to-image diffusion model adaptation.**  Text-to-image (T2I) diffusion model adaptation is also referred to *T2I personalization* or *subject-driven image generation*. This aims to adapt a given model to a *new concept* by providing a few images and binding the new concept to a unique token. As a result, the adaptation model can generate various renditions of the new concept guided by text prompts. Depending on whether the adaptation method is finetuning the T2I model, they are categorized into two main streams. One of the most representative methods focuses on learning new concept tokens while freezing the T2I generative backbones. Textual Inversion (TI) (Gal et al., 2023) is a pioneering work focusing on finding new pseudo-words by performing personalization in the text embedding space. The following works (Dong et al., 2022; Daras & Dimakis, 2022; Voynov et al., 2023; Han et al., 2023a) continue to improve this technique stream. Another stream is finetuning the T2I generative models while updating the modifier tokens. One of the most representative methods is DreamBooth (Ruiz et al., 2023a), where the pre-trained T2I model learns to bind a modified unique identifier to a specific subject given 3∼5 images, while it also updates the T2I model parameters. HyperDreamBooth (Ruiz et al., 2024) extends this approach for face domain by training per-subject LoRAs to inform a HyperNetwork that can rapidly adapt to new subjects. CAFE (Zhou et al., 2024) takes a different approach by leveraging instruction-based personalization through extensive datasets of image-instruction pairs. Custom Diffusion (Kumari et al., 2023) and other approaches (Han et al., 2023b; Chen et al., 2023b; Shi et al., 2023a) follow this pipeline and further improve the quality of the generation.

Finetuning methods often achieve state-of-the-art performance but introduce forgetting in large T2I models. While research focuses on improving new concept generation, it overlooks continuous model updating and forgetting mitigation. Recent works (Sun et al., 2024; Smith et al., 2023) address token forgetting but neglect other impacts of finetuning, such as semantic drifting in color, appearance, and visual recognition, which this paper explores.

**Assessing forgetting.**  The main challenge of continual learning is to learn incrementally and accumulate knowledge of new data while preventing *catastrophic forgetting*, which is defined as a sudden drop in performance on previously acquired knowledge (McCloskey & Cohen, 1989; McClelland et al., 1995). The vast majority of studies on continual learning focus on, what we here call, *closed-world forgetting*, where the knowledge of the network can be represented by its performance on a limited set of classes (Lopez-Paz & Ranzato, 2017; De Lange et al., 2021; Masana et al., 2022). However, as argued in the introduction, the growing importance of starting from large pretrained models (also known as foundation models), which have a vast prior knowledge, requires new techniques to assess forgetting. The forgetting of large language models (LLMs) during continual finetuning has received some attention in recent years, showing the importance of pretraining to mitigate forgetting (Cossu et al., 2024), however, they mainly evaluate on down-stream-task performance Scialom et al. (2022). To the best of our knowledge, *open-world forgetting* has not yet been systematically analyzed for text-based image generation models which multi-modal nature can further worsen the impact of forgetting due to misalignment of the modalities.

## 3 CUSTOMIZATION OF DIFFUSION MODELS

In this section, we briefly introduce text-to-image (T2I) models and the two main customization methods we will evaluate during our analysis. In addition, we introduce an alternative regularization method to further mitigate forgetting.

### 3.1 DIFFUSION MODELS

Diffusion models are a class of generative models that generate data by gradually denoising a sample from a pure noise distribution. The process is modeled in two stages: a forward process and a reverse process. In the forward process, Gaussian noise is iteratively added to data samples, typically over $T$ steps, forming a Markov chain. At each step, the transition is defined as:

$$q(\mathbf{x}_t|\mathbf{x}_{t-1}) = \mathcal{N}(\mathbf{x}_t; \sqrt{1-\beta_t}\mathbf{x}_{t-1}, \beta_t\mathbf{I}), \tag{1}$$

where $\beta_t$ controls the noise schedule.

The reverse process denoises the data by learning the conditional probability $p_\theta(\mathbf{x}_{t-1}|\mathbf{x}_t)$, typically parameterized by a neural network $\epsilon_\theta(\mathbf{x}_t, t)$ that predicts the added noise at each step. The model is trained by minimizing a simple mean-squared error between the true and predicted noise:

$$L(\theta) = \mathbb{E}_{t,\mathbf{x}_0,\epsilon} \left[ \|\epsilon - \epsilon_\theta(\mathbf{x}_t, t)\|^2 \right]. \tag{2}$$

Text-to-image diffusion models employ an additional conditioning vector $\mathbf{c} = \mathcal{E}(P)$ generated using a text encoder $\mathcal{E}$ and a text prompt $P$. These models have gained prominence for their ability to generate high-quality, diverse samples, often outperforming other generative models like GANs and VAEs in terms of mode coverage and sample quality (Ho et al., 2020; Song et al., 2021).

### 3.2 CUSTOMIZATION APPROACHES

Diffusion models often require finetuning for specific domains or user needs. This involves introducing new conditioning mechanisms or retraining on specialized datasets. This paper applies two adaptation methods to evaluate finetuning's impact on image generation models.

**Dreambooth** (Ruiz et al., 2023b) enables personalization of diffusion models by finetuning them with a small set of images. It reuses an infrequent token of the vocabulary to represent a unique subject, allowing the model to generate images of the subject in varied contexts or styles. This approach induces *language drift* and *reduced output diversity* in the model, which is mitigated by replaying class-specific instances alongside the subject training, called *prior preservation loss*. The final training objective reads

$$\mathbb{E}_{\epsilon,\mathbf{x},\mathbf{c},t}[w_t\|\epsilon - \epsilon_\theta(\mathbf{x}_t, \mathbf{c}, t)\| + \lambda w_{t'}\|\epsilon' - \epsilon_\theta(\mathbf{x}_{t'}^{\mathrm{pr}}, \mathbf{c}^{\mathrm{pr}}, t')\|], \tag{3}$$

where $\lambda$ is a weighting parameter, and $\mathbf{x}_t^{\mathrm{pr}}$ and $\mathbf{c}^{\mathrm{pr}}$ come from the prior dataset. DreamBooth is especially useful for personalized content generation where subject fidelity is critical.

**Custom diffusion** (Kumari et al., 2023) is another approach aimed at efficiently finetuning diffusion models with minimal data and compute. This method observes that the cross-attention layer parameters undergo the most change during personalization, so they propose to only update the key and value projections in these layers. It introduces a token into the text-encoder representing a unique subject, rather than reusing an old one. By freezing the majority of the model's parameters and focusing updates on a few key layers, Custom Diffusion facilitates rapid customization with less degradation in image quality. Prior preservation loss is maintained, since language drift is still experienced otherwise.

**Customized Model Set** In our experiments, we will evaluate Dreambooth and Custom Diffusion. We adapt both these models to ten different concepts based on 5 images per concept. The concepts are 'lamp', 'vase', 'person2','person3','cat','dog','lighthouse','waterfall','bike' and 'car' taken from CustomConcept101 (Kumari et al., 2023). We will refer to these ten models for both DreamBooth and Custom Diffusion as the *Customized Model Set*.

## 3.3 DRIFT CORRECTION

The two studied approaches, Dreambooth (Ruiz et al., 2023b) and Custom Diffusion (Kumari et al., 2023), apply finetuning to adapt to the new data: they mainly focus on how good the learned model is on the target data, and do not study the possible detrimental effects for other classes. The Dreambooth method includes a method called *prior regularization*, which by replaying general instances of the concept being learned (see Eq. 3), helps to prevent the model from overfitting to the new data and ensures that the representation of the superclass remains stable. This same mitigation strategy is also applied in custom diffusion (Kumari et al., 2023).

In this paper, we propose another regularization technique that can be applied during new concept learning. The method is remarkably simple and is motivated from continual learning literature. This field has proposed a variety of methods to counter forgetting during the learning of new concepts (De Lange et al., 2021). Regularization methods aim to regularize the learning of new concepts in such a way that it does not change weights which were found relevant for previous tasks. The field differentiates between parameter regularization methods, like EWC (Kirkpatrick et al., 2017) which directly learn an importance weight for all the network parameters, or functional (or data) regularization, like Learning-without-Forgetting (Li & Hoiem, 2017; Pan et al., 2020) which regularizes the weights indirectly by imposing a penalty on changes between the (intermediate) outputs of a previous and current model.

We propose to apply a functional regularization loss to the network during the training of new concepts. Our loss, called *drift correction loss*, constrains the difference between the outputs of the pre-trained and fine-tuned models when the new concept is not present in the prompt. It has the following form:

$$\mathbb{E}_{\epsilon, \mathbf{x}, \mathbf{c}, t}[w_t \| \epsilon - \epsilon_\theta(\mathbf{x}_t, \mathbf{c}, t) \| + \lambda w_{t'} \| \epsilon_{\theta*}(\mathbf{x}_{t'}^{\mathrm{pr}}, \mathbf{c}^{\mathrm{pr}}, t') - \epsilon_\theta(\mathbf{x}_{t'}^{\mathrm{pr}}, \mathbf{c}^{\mathrm{pr}}, t') \|], \tag{4}$$

where the second term is the distillation loss, $\lambda$ is a relative weighting parameter and $\epsilon_{\theta*}$ is the base model. This loss helps to maintain consistency in the model's internal representations while allowing it to learn new information effectively. For the training process, we choose instances from the same class as the concept being learned, similar to those used by prior regularization. The change between our proposed drift correction method (Eq. 4) and the existing prior regularization (Eq. 3) is that we do not require the finetuned network to estimate the true forward noise, but instead we want it to estimate the same noise as the original starting network. We will see that this small change significantly improves stability and mitigates forgetting.

In our evaluation, we provide results for *DreamBooth (DB)* which includes the prior regularization, for *Dreambooth with Drift Correction (DB-DC)* which also includes the prior regularization and for *DreamBooth with Drift Correction without the prior regularization (DB-DC\pr)*. Similarly, we show results for the various variants of Custom Diffusion (*CD, CD-DC*, and *CD-DC\pr*).

## 4 OPEN-WORLD FORGETTING IN GENERATIVE MODEL ADAPTATION

In this section, we explore the effects of finetuning on foundational image generation models, particularly how even slight modifications can significantly impair the model's ability to retain previously acquired knowledge. We hypothesize that this degradation affects not only the model's performance on newly introduced tasks, but also its capacity to accurately reproduce or classify previously learned concepts. Given the broad scope of knowledge encompassed by the pretrained model, we refer to this phenomenon as *open-world forgetting*.

As an initial experiment, to assess open-world forgetting, we evaluate both the original unaltered model (called *base model* from now on) and the *Customized Model Set* on 10,000 user prompts from DiffusionDB (Wang et al., 2023) dataset (prompt examples are provided in Appendix B.1). Specifically, we measure the change of the resulting images using the cosine distance between CLIP-I encodings (Radford et al., 2021) when generating images with the same prompt and seed. Distances in the CLIP-I embedding are related to semantic similarity between images, with smaller distances indicating more similar visual content and larger distances suggesting more significant differences in the generated images. The distribution is plotted in Figure 2. For a detailed description of our experimental setup, please refer to Appendix B.1. It is important to note that a personalization method that does not alter the model would yield identical image outputs, resulting in a plot density concentrated at 1.

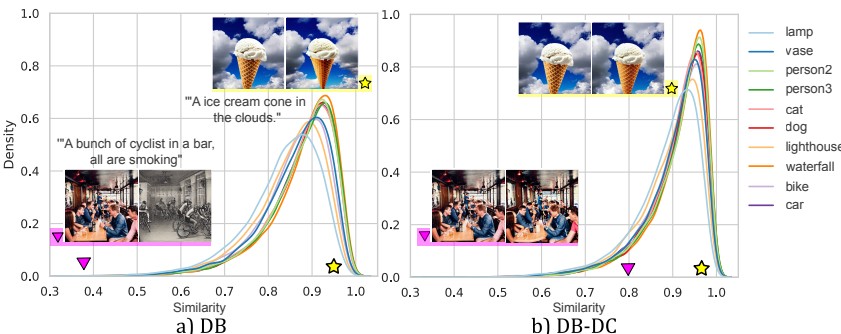

Figure 2: Similarity (measured as cosine distance in CLIP-I embedding space) between models before and after adaptation. Each curve represents one of the 10 models from the Customized Model Set. a) Results with DreamBooth adaptation (includes prior regularization). b) Results with DreamBooth with Drift Correction. For more results see Appendix A.

When considering Figure 2a, we observe that even though most of the prompts from DiffusionDB are not related with the selected trained concepts, there is a significant part of the distribution that is shifted to the left. This shows indeed that the representations of the original model have changed. Furthermore, further analysis shows that open-world forgetting significantly alters the output in different ways, as illustrated by the samples in Figure 2a. For instance, a sampled pair from the most dissimilar outputs (purple triangle) shows a complete change in content, colors, and scene composition that no longer matches the prompt. In contrast, a very similar pair (yellow star) closely adheres to the original model's output, with only changes in color or details. Interestingly, when looking at Figure 2b where we apply the proposed Drift Correction to DreamBooth, the distribution shifts to the right, showing that the drift has been reduced considerably.

To better assess the impact of open-world forgetting, we propose to categorize the effects into two distinct types: **semantic** and **appearance drift**. Semantic drift implies a change at the class or object level, where one concept is effectively misencoded as another. Appearance drift, on the other hand, refers to shifts in the appearance of a concept that do not necessarily imply a change in recognition (e.g., alterations in color, texture, or scene composition). It is important to note that these two categories are highly correlated, and changes in either of them impact the other.

## 4.1 SEMANTIC DRIFT

Semantic drift refers to alterations in a model's representation that cause the generation of semantically divergent content following customization. In the experiment depicted in Figure 2a, almost all prompts exhibit some level of drift, with a notable long tail of highly dissimilar generations. Many of these pronounced deviations have resulted in the generation of content that semantically no longer align with the input prompt.

To evaluate how semantic drift affects generative models, we use a straightforward approach: we utilize the model's internal representations on different classification tasks (Mittal et al., 2023; Tang et al., 2023). It is based on a recent insight that showed that diffusion models can be directly applied for zero-class classification, by leveraging the conditional likelihood estimation of the model. Concretely, we use Diffusion Classifier (Li et al., 2023), where a posterior distribution over classes $\{\mathbf{c}_i\}_{i=1}^N$ is calculated as:

$$p_\theta(\mathbf{c}_i \mid \mathbf{x}) = \frac{\exp\{-\mathbb{E}_{t,\epsilon}[\|\epsilon - \epsilon_\theta(\mathbf{x}_t, \mathbf{c}_i)\|^2]\}}{\sum_{j=1}^N \exp\{-\mathbb{E}_{t,\epsilon}[\|\epsilon - \epsilon_\theta(\mathbf{x}_t, \mathbf{c}_j)\|^2]\}}. \tag{5}$$

Monte Carlo sampling is performed over $t_i$ and $\epsilon$ to obtain a classifier from the model $\epsilon_\theta$.

While this method offers a simple, parameter-free approach to evaluating semantic drift, it is worth noting that alternative techniques have been proposed to assess the representation space of diffusion models. These include linear probing on activations (Xiang et al., 2023), analysis of hierarchical features (Mukhopadhyay et al., 2023), and methods requiring a preliminary likelihood maximization

Table 1: Average zero-shot classification using the T2I models of the Customized Model Set for several image classification datasets. Worst class drop between parenthesis.

|  | CIFAR10 | STL10 | Flowers | Pets | ObjectNet | Food | Aircraft |
|---|---|---|---|---|---|---|---|
| Base Model | 81.60 | 93.00 | 50.00 | 86.87 | 28.50 | 71.09 | 23.40 |
| DB | 75.92 (32.40) | 91.30 (18.60) | 46.61 (64.00) | 82.61 (36.43) | 25.26 (56.00) | 65.48 (56.00) | 19.36 (58.00) |
| DB-DC | 80.98 (14.00) | 93.36 (4.40) | 49.29 (42.00) | 86.64 (17.14) | 27.72 (42.00) | 69.07 (44.00) | 21.42 (48.00) |
| DB-DC\pr | 80.60 (14.00) | 92.94 (5.20) | 49.06 (40.00) | 86.37 (16.43) | 27.45 (46.00) | 68.79 (44.00) | 21.54 (44.00) |
| CD | 79.98 (17.00) | 91.40 (12.20) | 47.65 (66.00) | 83.46 (33.57) | 25.75 (58.00) | 65.25 (56.00) | 19.44 (58.00) |
| CD-DC | 82.36 (9.00) | 93.02 (5.00) | 49.33 (42.00) | 86.37 (16.43) | 27.91 (42.00) | 69.19 (44.00) | 21.94 (46.00) |
| CD-DC\pr | 82.04 (10.80) | 92.76 (6.00) | 49.16 (44.00) | 86.70 (20.00) | 27.77 (42.00) | 68.99 (44.00) | 21.56 (48.00) |

stage (Chen et al., 2023a). However, these alternatives often involve additional computational steps or are subject to specific settings, potentially limiting their applicability or introducing complexity to the evaluation process.

We conducted zero-shot classification experiments across multiple datasets spanning diverse domains to quantify the semantic drift of the models. We perform two measurements. First, we measure the *average zero-shot classification score* for the various models (the results are averaged over the 10 models of the Customized Model Set). Second, we establish the performance of the original pretrained model as the baseline, and measure the presence of semantic drift by calculating the drop in accuracy from the baseline. We also report the *worst class drop* which is the drop in accuracy of the class that has suffered the largest deterioration due to the adaptation. For further details on the classification method, please refer to Appendix B.2.

Table 2: Concept fidelity (DINO, CLIP-I) and prompt fidelity (CLIP-T). Drift Correction maintains fidelity across metrics.

|  | DINO | CLIP-I | CLIP-T |
|---|---|---|---|
| DB | 0.42 | 0.68 | 0.79 |
| DB-DC | 0.43 | 0.68 | 0.78 |
| DB-DC\pr | 0.43 | 0.68 | 0.78 |
| CD | 0.44 | 0.69 | 0.79 |
| CD-DC | 0.44 | 0.69 | 0.79 |
| CD-DC\pr | 0.44 | 0.69 | 0.79 |

The results in Table 1 are surprising, average zero-shot classification accuracy drop significantly on all the datasets: adapting a huge generative image foundation model to just five images of a new concept has a vast impact throughout the latent space of the diffusion models. When applying DreamBooth, average zero-shot performance drops by over 4% on CIFAR10, Pets, Food and Aircraft. If we look at individual classes, the impact can be much larger. As indicated by the *worst class drop*, for some classes, zero-shot performance drops by over 60% (e.g. 'vacuum cleaner' gets recognized as 'microwave', 'drill' or 'laptop'). We show that these drops in performance are mitigated to a large extent by our alternative Drift Correction results (see DB-DC and CD-DC results) and their average zero-shot classification scores are in general within 1% of the base model. Removing the prior regularization from our method (see DB-DC\pr and CD-DC\pr) leads to only slightly lower results, showing the impact of our proposed regularization method. Also, worst class drop significantly reduces when applying DC, but for some datasets remains still high.

We employ three primary metrics to assess image generation quality. CLIP-I is calculated as the average pairwise cosine similarity between CLIP (Radford et al., 2021) embeddings of real and generated images. DINO uses the same pairwise cosine similarity but with DINO (Caron et al., 2021) ViT-S16 embeddings. This metric is preferred over CLIP-I as it does not ignore differences between subjects of the same class. CLIP-T measures the CLIP embedding cosine similarity between the prompt and the generated image, and is used to evaluate prompt fidelity. In Table 2 we can see that the proposed regularization method DC does not negatively impact the image generation quality of the learned concepts.[1]

## 4.2 Appearance Drift

While open-world forgetting does not always result in significant changes to the core content of the image, as shown in Figure 2a, it notably affects intra-class variation, color distribution, and texture

---

[1]The results with standard deviations for Table 1 and 2 are provided in Appendix B.

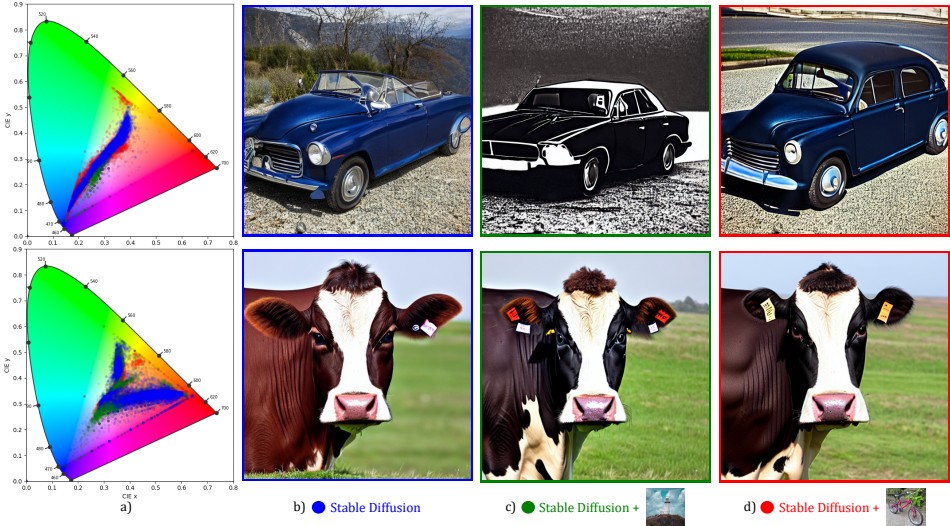

Figure 3: Appearance drift as consequence of DreamBooth customization. **a)** chromaticity plot of pixels of three realization of the prompts ('photo of a car/cow') and the same seed with different models, namely **b)** the base model, **c)** model adapted to lighthouse and d) model adapted to bike.

characteristics. We define these collective changes as *appearance drift*, a phenomenon that alters the model's representation space in subtle yet impactful ways. Figure 3 demonstrates two key aspects of appearance drift; intra-class and contextual variation (first row), where different customizations of the base model lead to changes in car brand and background, while maintaining the overall concept of 'car'. Color shift (second row), where the color palette of the generated images changes significantly, even when the intra-class characteristics and background remain relatively constant.

Appearance drift manifests through alterations in visual attributes at varying degrees of intensity. Finetuning can cause the model to reinterpret these visual features, leading to inconsistencies between original and newly generated outputs. Although initially subtle, appearance drift can substantially impact customized models. For example, when attempting to learn and generate a set of new concepts within the same context (e.g., for synthetic dataset creation or advertising purposes), each customization of the base model may result in color and content changes. This variability makes it challenging to achieve consistent results across multiple iterations. Moreover, as the customization process alters the model's manifold, the resulting model becomes less reliable in domains outside the scope of the customization training images. This limitation highlights the importance of understanding and mitigating appearance drift in applications of fine-tuned text-to-image models.

**How to measure appearance drift?**   Quantifying appearance drift presents unique challenges due to the inherent variability in text-to-image model outputs. Traditional metrics like LPIPS (Zhang et al., 2018) and DIST (Ding et al., 2020) are designed for image pair comparisons. However, the inherent variability in T2I model outputs means that images generated from the same prompt can vary significantly due to changes in seed, model weights, and prompt interpretation. Comparing just two images fails to capture the full range of possible outputs and does not adequately represent the model's capabilities or biases. Consequently, conclusions drawn from such limited comparisons may lack statistical significance.

To address this variability, the research community has employed metrics that measure distances between probability distributions of real-world observations and generated data[2]. For example, FID (Heusel et al., 2017) assumes Gaussian distributions and compute its distance. KID (Bińkowski et al., 2018) is similar but uses a kernel-based approach that is more reliable. In addition to these metrics, we also propose a new metric that directly measures the color drift between image sets.

---

[2]In general FID and KID require a set of real images for comparison. In our study, we consider the images generated by the original model as the "real" set, as we are measuring the shift from this initial distribution.

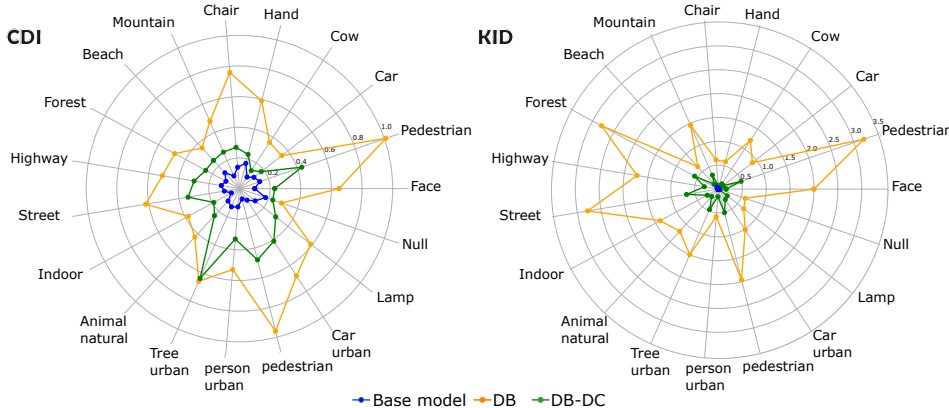

Figure 4: Appearance drift as consequence of customization measured with (left) Color Drift Index (CDI) and (right) Kernel Inception Distance (KID). The orange and green line represent the distance between the base model and the customized model. The blue line is a control line, representing the distance between two sets of images generated from different seeds both with the base model. Lines close to the origin are better.

**Color Drift Index.** With the aim to introduce a clearly interpretable difference measure for the color content of the generated images, we propose a novel approach that measures specific properties in pixel space. Our method focuses on color assessment, as traditional image generation metrics tend to be more sensitive to intra-class and texture variations. Inspired by natural image statistics, we introduce the Color Drift Index (CDI), which estimates the color distribution associated with a specific concept by analyzing a large number of generated images.

We utilize the CIE chromaticity diagram[3], where each pixel color is projected onto a lobe-shaped space representing all visible colors. Given a set of images $I = \{\mathbf{x}_i\}$ and their density distribution in the CIE chromaticity diagram $p^{\text{CIE}}(I)$, we calculate the CDI as the Wasserstein distance (Panaretos & Zemel, 2019) between the color distributions of two sets of images:

$$\text{CDI}(I_a, I_b) = W_p(p^{\text{CIE}}(I_a), p^{\text{CIE}}(I_b)). \tag{6}$$

We evaluate the appearance drift using the CDI together with KID (FID results are presented in Appendix B.3). We conducted a comprehensive experiment adopting the carefully curated selection of common concepts from the dataset of Torralba & Oliva (2003). For each prompt, we generated 1,000 images using both original and the ten models from the Customized Model Set. Figure 4 presents the mean values of the metrics across several adaptations, providing a visual representation of the differences captured by each measure. For a detailed overview of the results, including individual model performances, refer to Table 8 in the appendix. To help interpret the KID and CDI scores, we provide a control setting (blue line). In this configuration, we measure CDI and KID between images generated with the base model but using different seeds (functioning as a lower bound). If we are sampling from the same distribution, the *base model* should yield lower distances (approaching zero as the number of samples grows) than the customized models (DB and CD).

The results in Figure 4 reveal two key insights. First, the *base model* consistently produces smaller distance values compared to DB, confirming that the distribution of each concept is indeed changing due to appearance drift. Second, each concept is affected differently by the drift, attributable to the fact that each concept relates to different parts of the model's manifold. Furthermore, as demonstrated in Appendix C, the magnitude and nature of the drift vary as a function of the content in the replay buffer and training images. Also, importantly, Figure 4 shows that our proposed method (DC) significantly reduces the impact of the appearance drift introduced by customization methods. Especially, the drift measure in KID is considerably reduced.

These findings underscore the complexity and pervasiveness of appearance drift in fine-tuned text-to-image models. They highlight the need for robust mitigation strategies and careful consideration when deploying customized models in real-world applications, emphasizing the importance of ongoing research in this area to ensure the reliability and consistency of generated outputs.

---

[3]Our approach offers the added benefit of being applicable across multiple color spaces.

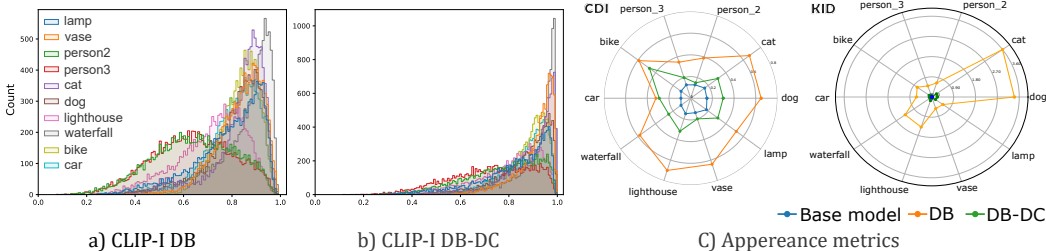

a) CLIP-I DB      b) CLIP-I DB-DC      C) Appereance metrics

Figure 5: Similarity (measured as cosine distance in CLIP-I embedding space) and perceptual metrics between models before and after adaptation. For each concept trained, we evaluate closely related concepts to measure the local drift. **a)** Results with DreamBooth adaptation (includes prior regularization). **b)** Results with DreamBooth with Drift Correction. **c)** Color Drift Index (CDI) and Kernel Inception Distance (KID). For more results see Appendix A.

### 4.3 LOCAL DRIFT

In this paper, we have focused on drift throughout the whole diffusion model manifold. Previous works, especially those in the machine unlearning community (Gandikota et al., 2023), have concentrated on *local drift*. When removing a concept from a model, it is believed to mainly impact the representation of closely related concepts (hence the name local drift). Our findings suggest that the effects of finetuning are more pervasive than previously thought, potentially influencing the model's understanding and representation of far-away categories as well as close-by (local) categories.

Here, we repeated our experiments from Section 4.1 and 4.2 to measure the local semantic and local appearance drift. For this setup, we generated 1,000 samples of the closest concepts (superclasses) to each trained model (see Appendix B.1 for the details) and evaluated the CLIP-I, CDI, FID, and KID metrics. As Figure 5a shows, the semantic drift is showing a significant shift towards the left, indicating that local drift is more pronounced. For appearance drift, Figure 5c depicts a more uniform color and KID shift over all the models; this shows that related concepts are affected with a similar magnitude by appearance drift. Again the application of our proposed Drift Correction method greatly reduces both the local semantic drift (as measured in Figure 5b and it almost removes the local appearance drift as measured by KID, even though some color drift remains (see Figure 5c).

### 5 DISCUSSION AND CONCLUSION

Our investigation into unintended consequences of generative model adaptation reveals several key findings. First, we demonstrate that finetuning foundational generative models leads to substantial *open-world forgetting*, manifesting as both *semantic* and *appearance drift*. Our results show that even minor adaptations can cause significant deterioration in the model's ability to maintain its original capabilities across a broad spectrum of concepts and visual attributes. To quantify these effects, we introduced novel evaluation approaches: measuring semantic drift through zero-shot classification performance across diverse image classification tasks, and assessing appearance drift through our proposed Color Drift Index combined with traditional metrics like KID. These methods provide a framework for understanding and measuring the impact of model adaptation on both semantic understanding and visual representation. Additionally, we propose a technique to mitigate open-world forgetting using functional regularization. Our experiments demonstrate that this method effectively preserves foundational knowledge while allowing for successful customization, offering a promising direction for developing more robust adaptation techniques.

The increasing proliferation of foundation models and their widespread adaptation across various domains underscores the importance of understanding and addressing open-world forgetting. While our study provides valuable insights and measurement techniques, the vast knowledge space of foundation models makes comprehensive evaluation challenging. Future research directions might explore active optimization methods to identify the most affected areas of model knowledge during adaptation. Furthermore, extending our methodology to other forms of model adaptation, such as unlearning techniques, remains an important area for future work.

ETHICAL STATEMENT

We acknowledge the potential ethical implications of deploying generative models, including issues related to privacy, data misuse, and the propagation of biases. All models used in this paper are publicly available, as well as the base training scripts. We will release the modified codes to reproduce the results of this paper. We also want to point out the potential role of customization approaches in the generation of fake news, and we encourage and support responsible usage of these models. Finally, we think that awareness of open-world forgetting can contribute to safer models in the future, since it encourages a more thorough investigation into the unpredictable changes occurring when adapting models to new data.

REPRODUCIBILITY STATEMENT

To facilitate reproducibility, we will make the entire source code and scripts needed to replicate all results presented in this paper available after the peer review period. We will release the code for the novel color metric we have introduced. We conducted all experiments using publicly accessible datasets. Elaborate details of all experiments have been provided in the Appendices.

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

# A QUALITATIVE RESULTS

## A.1 OPEN-WORLD FORGETTING

In Figure 6, we present more samples of generated images with appearance drift.

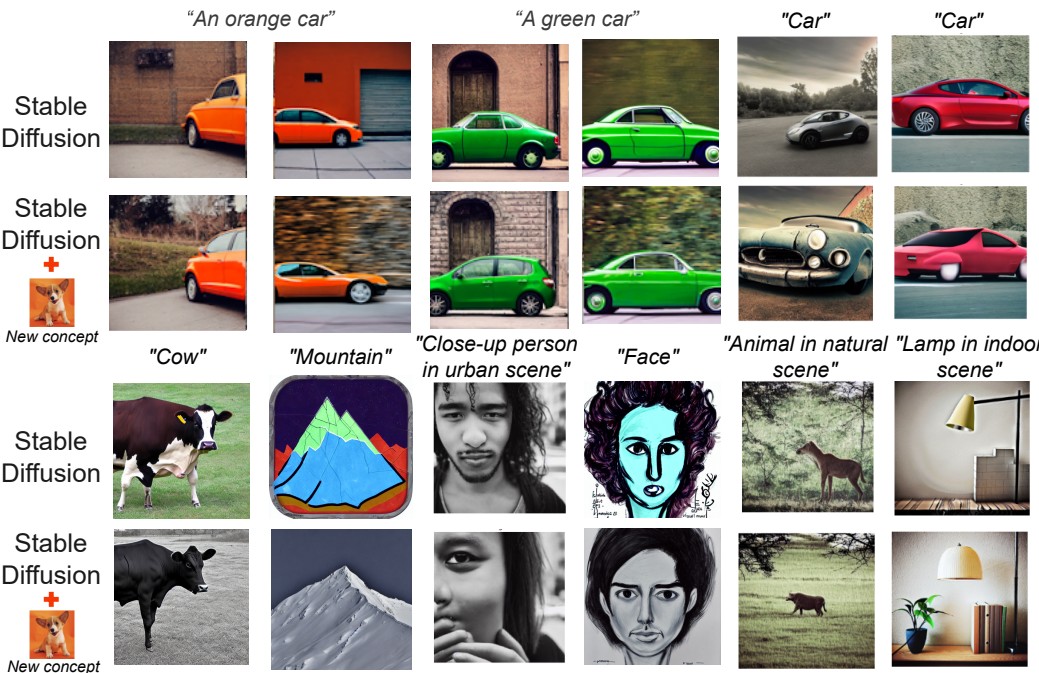

Figure 6: Several examples of appearance drift with DreamBooth. Images generated from the same initial seed.

## A.2 LOCAL DRIFT

In Figure 7 we present examples of local drift for the concept "dog".

## A.3 COMPARISON WITH DRIFT CORRECTION

To demonstrate the effectiveness of our proposed correction method, we provide visual examples showing how the pretrained model experiences semantic and appearance drifting, and how our method mitigates these issues. The comparative results are presented in Figures 8 and 9. These examples clearly illustrate both types of drifting in the baseline model and the improvements achieved through our correction approach.

## A.4 USER STUDY SAMPLES

We conducted a user study where participants were presented with image triads, as shown in Figure 10. Each triad consisted of a reference image in the center and two comparison images (labeled A and B) on either side. The methods evaluated were DreamBooth and Custom Diffusion and its corresponding Drift Corrected versions. Participants were given the following instructions:

*"Look at the three images shown: one in the center, and two options (A and B) on the sides. Your task is to determine which side image (A or B) is more visually similar to the center image."*

Stable Diffusion          Stable Diffusion ✚

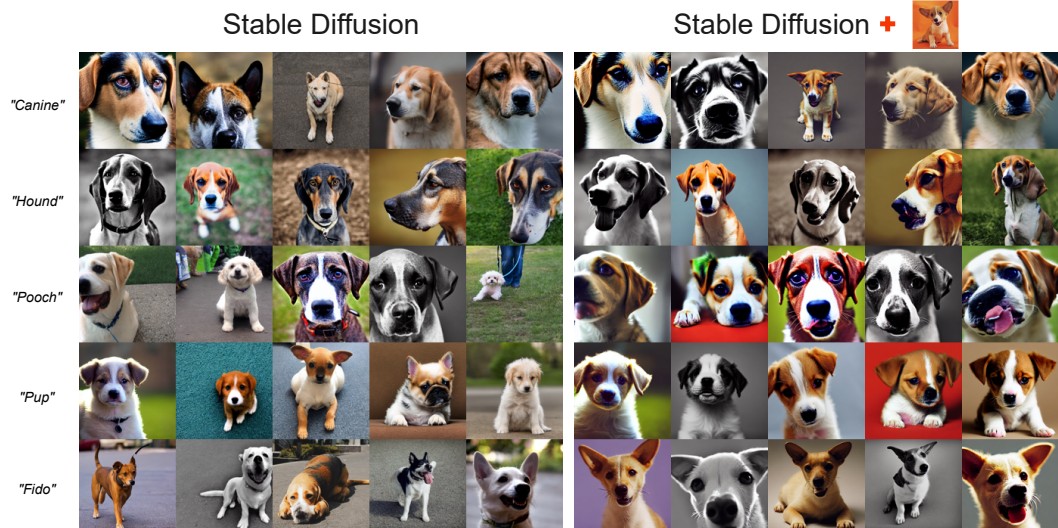

Figure 7: Several examples of local drift with DreamBooth. Images generated from the same initial seed. Note that the variety in viewpoint and breeds reduces significantly.

## B  EXPERIMENT DETAILS

This section outlines our experimental setup, including datasets, metrics, and training configurations.

### B.1  SEMANTIC DRIFT EVALUATION

**Datasets.**    To evaluate open-world forgetting, we select a random subset of 10,000 user prompts from DiffusionDB 2M (Wang et al., 2023) (Table 3). For adaptation training and evaluation, we choose a subset of 10 concepts from CustomConcept101 (Kumari et al., 2023), namely *decoritems_vase2, decoritems_lamp1, person_2, person_3, pet_cat5, pet_dog4, transport_bike, transport_car2, scene_lighthouse, scene_waterfall*. Each concept contains approximately 3-5 images. For superclass evaluation, we create a dataset of 10 synonyms with respect to each concept, which can be found in Table 4.

Table 3: DiffusionDB subset sample prompts. Shorter prompts selected for visualization purposes.

| DiffusionDB prompts |
|---|
| "*dafne keen, mad max, cinematic shot, 8k resolution*" |
| "*creepy horror movie characters, fog, rain, volumetric lighting, beautiful, golden hour, sharp focus, highly detailed, cgsociety*" |
| "*the railroad is a place of death. it's where the forgotten and the damned go to die. it's a place of dark secrets and hidden terror. photorealistic*" |
| "*samurai jack johnny bravo by salvador dali*" |
| "*Film still of Emma Watson as Princess Leia in Star Wars (1977)*" |
| "*a detailed figure of indigo montoya, first 4 figures, detailed product photo*" |
| "*a hyper scary pokemon, horror, creepy, big budget horror movie, by zdzisław beksinski, by dorian cleavenger*" |
| "*the war between worlds extremely detailed claymation art, dark, moody, foggy*" |
| "*a painting of Hatsune Miku by H. R. Giger, highly detailed, 4k digital art*" |
| "*a redneck with wings and horns wearing sunglasses and snake skin smoking a blunt, detailed, 4 k, realistic, picture*" |
| "*fantasy art 4 k ultra detailed photo caricature walter matthau as an fighter pilot*" |
| "*CG Homer Simpson as Thanos, cinematic, 4K*" |
| "*Full body portrait of Raven from Teen Titans (2003), digital art by Sakimichan, trending on ArtStation*" |
| "*bigfoot walking down the street in downtown Bremerton Washington*" |
| "*garden layout rendering with flowers and plants native to ottawa canada*" |
| "*a beautiful planet of guangzhou travel place of interest, chill time. good view, exciting honor. by david inshaw*" |
| "*an oil painting of Dwayne Johnson instead of Mona Lisa in the famous painting The Joconde painted by Leonardo Da Vinci*" |
| "*film still of danny devito as mario in live action super mario bros movie, 4 k*" |
| "*a beautiful artist's rendition of what the stable diffusion algorithm dreams about*" |

**Metrics.**    We employ three primary metrics to assess image generation quality. CLIP-I is calculated as the average pairwise cosine similarity between CLIP (Radford et al., 2021) embeddings of real and generated images. DINO uses the same pairwise cosine similarity method but with DINO

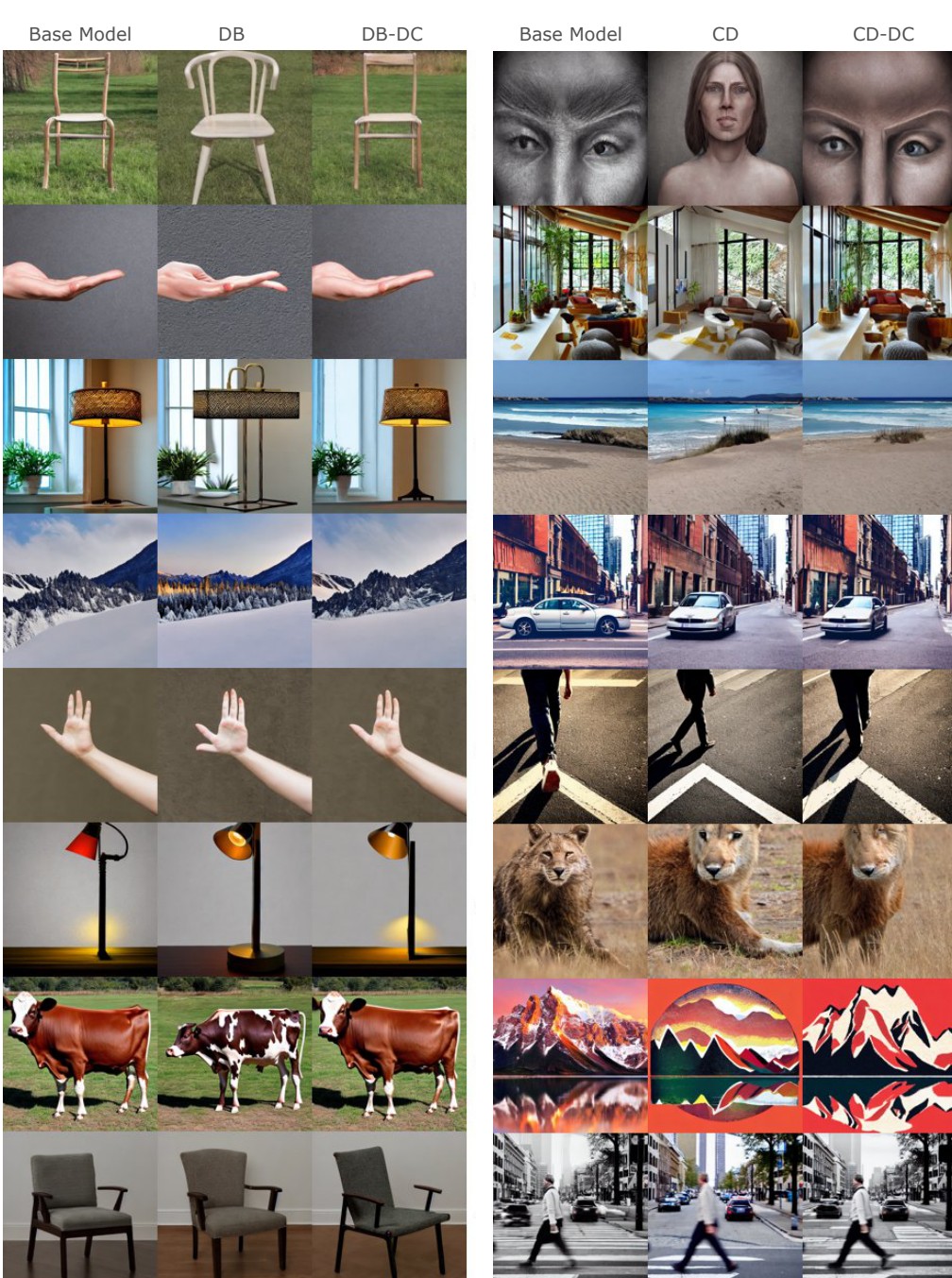

Figure 8: Qualitative result comparisons on diverse prompts for the pretrained model, a customization method and the proposed Drift Correction. These are random results and all generated from the same initial seed.

(Caron et al., 2021) ViT-S16 embeddings. This metric is preferred over CLIP-I as it does not ignore differences between subjects of the same class. CLIP-T measures the CLIP embedding cosine similarity between the prompt and the generated image, and is used to evaluate prompt fidelity.

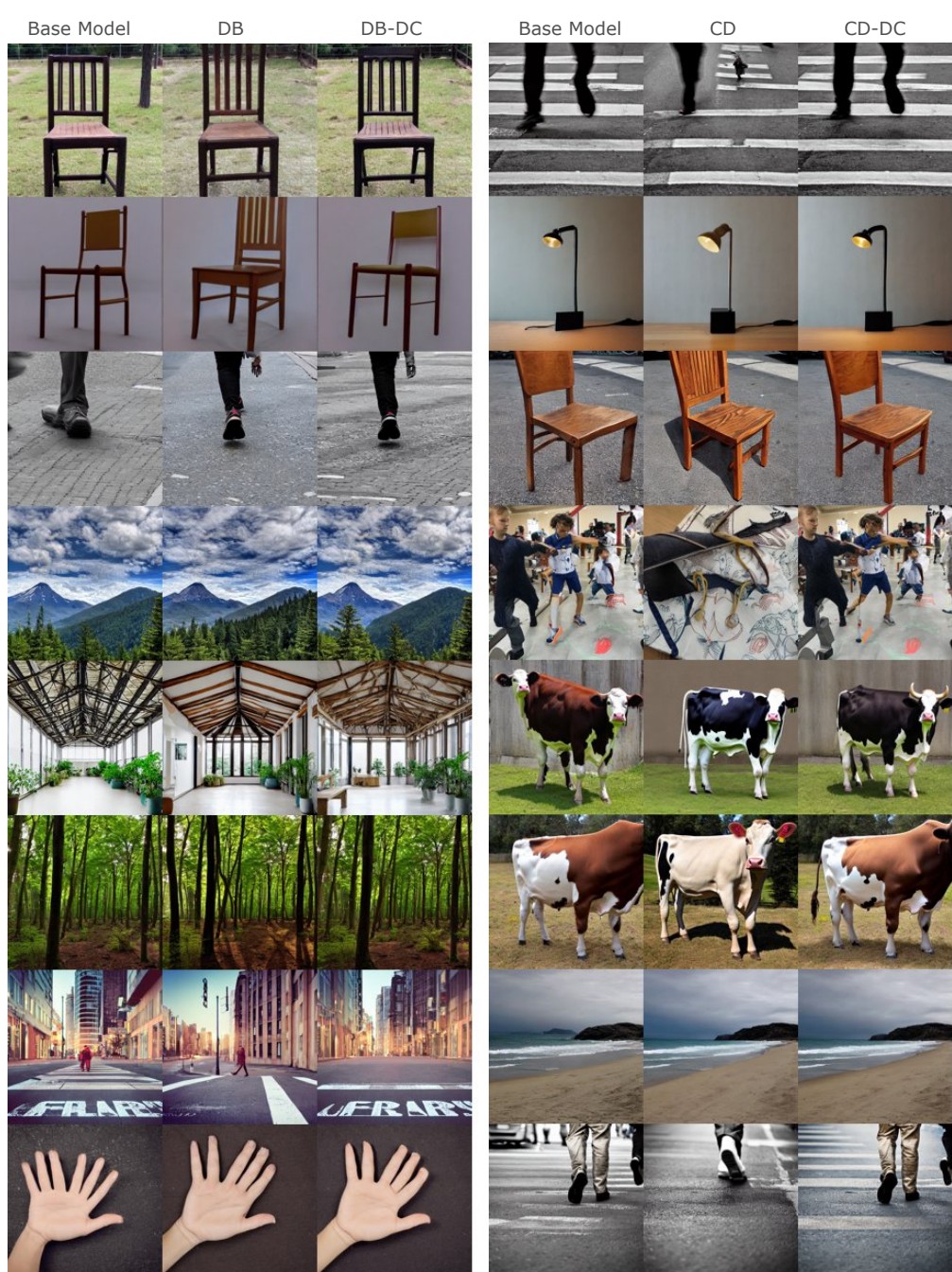

Figure 9: More qualitative result comparisons on diverse prompts for the pretrained model, a customization method and the proposed Drift Correction. These are random results and all generated from the same initial seed.

**Training configuration.** We adapt models using publicly available scripts from Diffusers (von Platen et al., 2022) for Dreambooth[4] and Custom Diffusion[5] applied to Stable Diffusion v1.5 (Rom-

---

[4] https://github.com/huggingface/diffusers/blob/main/examples/dreambooth/train_dreambooth_lora.py

[5] https://github.com/huggingface/diffusers/blob/main/examples/custom_diffusion/train_custom_diffusion.py

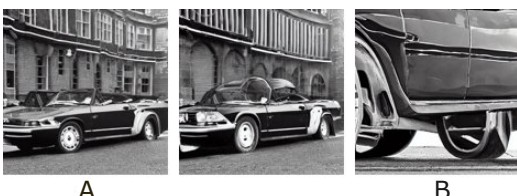

|  | A |  | B |
|---|---|---|---|

Figure 10: Example triad presented in the user study.

Table 4: Concept synonyms.

| Concept | Synonyms |
|---|---|
| bike | pedal cycle, velociped, roadster, bicycle, push bike, pushbike, cycle,wheels, two-wheeler, pedal bike |
| car | jalopy, ride, auto, vehicle, coupe, wheels, automobile, sedan, hatchback, motocar |
| cat | feline, grimalkin, mouser, moggy, tabby, puss, kitty, kitten, pussycat, tomcat |
| dog | canine, hound, pup, pooch, fido, puppy, mutt, man's best friend, doggy, cur |
| lamp | fixture, chandelier, light, illuminator, lantern, luminaire, glow, torch, sconce, beacon |
| lighthouse | light, coastal beacon, navigation light, pharos, seamark, watchover, beacon, guide light, light station, signal tower |
| person | gent, bloke, chap, gentleman, lad, guy, male, bro, fellow, dude |
| vase | urn, amphora, container, pitcher, carafe, receptacle, jar, vessel, pot, jug |
| waterfall | rapids, torrent, flume, cascade, spillway, cataract, plunge, chute, falls, deluge |

bach et al., 2022). Both methods use prior regularization unless otherwise stated, which is designed to prevent drifting towards the training concept. The set of images for prior regularization is generated from the base model before training starts. We use the LoRA script versions and refer to the resulting models as DB for DreamBooth and CD for Custom Diffusion. Full finetuning models exhibit the same or worse shortcomings as the LoRA models analyzed and are termed FT for fine-tuning, such as DB FT in Table 6. Since DB and CD are similar and to ensure a fair comparison, both methods use similar training settings: a learning rate of 1e-4, batch size of 1, 500 training steps, and no augmentations. The prior regularization uses a weighting of 1 and comprises 200 samples of generated images with the prompt "{concept}" for each concept, using default generation settings. For the drift correction method described in Section 3.3, all settings remain the same, and the weighting parameter is set to $\lambda = 10$.

## B.2 DIFFUSION CLASSIFIER

We employ the official released code of the Diffusion Classifier method[6]. However, due to computational constraints, we modify some parameters for our explorations. We reduce the keep list to $(10, 100)$ across all datasets while maintaining the trial list at $(5, 1)$. This significantly reduces computational time while resulting in minimal percentual score uncertainty. Additionally, datasets with many classes or samples were reduced to have a total number of samples of roughly 500 by random selection of the samples of each class. The datasets configuration can be seen in Table 5. It is worth noting that the original ObjectNet has 313 classes, but Diffusion Classifier only uses 113 for testing. We also use a fixed noise for consistent evaluations.

Table 5: Dataset configurations to evaluate Diffusion Classifier.

| Dataset | Food | CIFAR10 | Aircraft | Pets | Flowers | STL10 | ObjectNet |
|---|---|---|---|---|---|---|---|
| # classes | 101 | 10 | 100 | 37 | 102 | 10 | 113 |
| Samples / class | 5 | 50 | 5 | 14 | 5 | 50 | 5 |
| Total samples | 505 | 500 | 500 | 518 | 510 | 500 | 565 |

The standard deviations from Table 1 and 2 are included in Table 6 and 7, respectively.

As further illustration in Figure 11 of the zero-shot classification accuracy, we provide the results for one of the models, namely "decoritems_lamp1" for all the datasets for DreamBooth. We can observe

---

[6]https://github.com/diffusion-classifier/diffusion-classifier

Table 6: Zero-shot classification using the T2I model. Personalized models suffer from degraded representations. Worst class drop between parenthesis. Scores with standard deviation across models.

|  | Food | CIFAR10 | Aircraft | Pets | Flowers | STL10 | ObjectNet |
|---|---|---|---|---|---|---|---|
| Base Model | 71.09 | 81.60 | 23.40 | 86.87 | 50.00 | 93.00 | 28.50 |
| DB FT | $61.50_{\pm5.95}$ | $69.86_{\pm6.79}$ | $16.04_{\pm3.81}$ | $79.11_{\pm5.29}$ | $43.18_{\pm6.58}$ | $87.04_{\pm4.63}$ | $21.52_{\pm3.41}$ |
| DB | $65.48_{\pm2.59}$ | $75.92_{\pm5.81}$ | $19.36_{\pm2.69}$ | $82.61_{\pm3.33}$ | $46.61_{\pm2.49}$ | $91.30_{\pm2.05}$ | $25.26_{\pm1.85}$ |
| DB-DC | $69.07_{\pm1.58}$ | $80.98_{\pm2.57}$ | $21.42_{\pm0.45}$ | $86.64_{\pm0.92}$ | $49.29_{\pm1.33}$ | $93.36_{\pm0.70}$ | $27.72_{\pm1.56}$ |
| DB-DC\pr | $68.79_{\pm1.78}$ | $80.60_{\pm2.46}$ | $21.54_{\pm0.77}$ | $86.37_{\pm0.78}$ | $49.06_{\pm1.12}$ | $92.94_{\pm0.95}$ | $27.45_{\pm1.15}$ |
| CD | $65.25_{\pm2.76}$ | $79.98_{\pm4.21}$ | $19.44_{\pm1.97}$ | $83.46_{\pm2.97}$ | $47.65_{\pm2.34}$ | $91.40_{\pm1.60}$ | $25.75_{\pm2.16}$ |
| CD-DC | $69.19_{\pm1.73}$ | $82.36_{\pm1.91}$ | $21.94_{\pm1.51}$ | $86.37_{\pm1.28}$ | $49.33_{\pm1.16}$ | $93.02_{\pm0.97}$ | $27.91_{\pm1.30}$ |
| CD-DC\pr | $68.99_{\pm1.73}$ | $82.04_{\pm2.23}$ | $21.56_{\pm1.94}$ | $86.70_{\pm1.22}$ | $49.16_{\pm1.31}$ | $92.76_{\pm0.85}$ | $27.77_{\pm1.58}$ |

Table 7: Concept fidelity (DINO, CLIP-I) and prompt fidelity (CLIP-T). Drift Correction maintains fidelity across metrics.concept evaluation

|  | DINO | CLIP-I | CLIP-T |
|---|---|---|---|
| DB | $0.4241_{\pm0.1503}$ | $0.6764_{\pm0.1046}$ | $0.7896_{\pm0.0296}$ |
| DB-DC | $0.4283_{\pm0.1584}$ | $0.6817_{\pm0.1097}$ | $0.7799_{\pm0.0324}$ |
| DB-DC\pr | $0.4315_{\pm0.1585}$ | $0.6841_{\pm0.1086}$ | $0.7776_{\pm0.0322}$ |
| CD | $0.4422_{\pm0.1378}$ | $0.6934_{\pm0.0902}$ | $0.7916_{\pm0.0266}$ |
| CD-DC | $0.4381_{\pm0.4381}$ | $0.6925_{\pm0.0888}$ | $0.7899_{\pm0.0280}$ |
| CD-DC\pr | $0.4382_{\pm0.1351}$ | $0.6935_{\pm0.0872}$ | $0.7872_{\pm0.0264}$ |

that the adaptation leads to drops on most classes (identified in red) but can also occasionally result in a performance increase (in green).

## B.3 APPEARANCE DRIFT FULL TABLES

See Table 8 for the full results which have been used for the generation of Figure 4. The prompt are '0:Face','1:Pedestrian','2:Car','3:Cow','4:Hand','5:Chair','6:Mountain', '7:Beach','8:Forest','9:Highway','10:Street','11:Indoor', '12:Animal in natural scene', '13:Tree in urban scene', '14:Close-up person in urban scene','15:Far pedestrian in urban scene','16:Car in urban scene','17:Lamp in indoor scene', '18:empty prompt'.

| Prompt | Vanilla | | | DB | | | DB-DC | | | CD | | | CD-DC | | |
|---|---|---|---|---|---|---|---|---|---|---|---|---|---|---|---|
|  | CDI | FID | KID | CDI | FID | KID | CDI | FID | KID | CDI | FID | KID | CDI | FID | KID |
| 00 | 0.10 | 24.75 | 0.00 | 0.65 | 43.74 | 2.00 | 0.23 | 21.70 | 0.16 | 0.48 | 37.01 | 1.22 | 0.24 | 24.90 | 0.29 |
| 01 | 0.14 | 44.55 | 0.00 | 1.01 | 79.78 | 3.22 | 0.43 | 46.16 | 0.50 | 0.82 | 72.44 | 2.63 | 0.33 | 51.68 | 0.79 |
| 02 | 0.12 | 19.70 | 0.01 | 0.35 | 28.31 | 0.90 | 0.18 | 18.37 | 0.12 | 0.31 | 27.14 | 0.58 | 0.19 | 20.11 | 0.15 |
| 03 | 0.09 | 29.17 | 0.01 | 0.36 | 41.81 | 1.22 | 0.14 | 27.40 | 0.13 | 0.40 | 42.25 | 1.13 | 0.23 | 31.62 | 0.37 |
| 04 | 0.17 | 35.93 | 0.01 | 0.59 | 40.77 | 0.59 | 0.23 | 30.53 | 0.07 | 0.57 | 42.65 | 0.70 | 0.32 | 33.76 | 0.22 |
| 05 | 0.14 | 19.65 | 0.01 | 0.76 | 28.15 | 0.61 | 0.27 | 17.93 | 0.08 | 0.58 | 26.45 | 0.57 | 0.34 | 20.09 | 0.17 |
| 06 | 0.09 | 34.22 | 0.00 | 0.48 | 52.80 | 1.46 | 0.26 | 34.55 | 0.32 | 0.57 | 60.90 | 2.16 | 0.34 | 41.52 | 0.72 |
| 07 | 0.14 | 34.89 | 0.05 | 0.36 | 42.58 | 0.64 | 0.25 | 29.45 | 0.14 | 0.41 | 48.98 | 1.19 | 0.22 | 33.66 | 0.26 |
| 08 | 0.10 | 25.50 | 0.00 | 0.48 | 52.42 | 2.79 | 0.25 | 26.78 | 0.57 | 0.42 | 55.33 | 2.95 | 0.29 | 32.76 | 1.01 |
| 09 | 0.12 | 27.51 | 0.01 | 0.51 | 41.82 | 1.73 | 0.30 | 28.22 | 0.30 | 0.43 | 41.37 | 1.40 | 0.33 | 27.80 | 0.52 |
| 10 | 0.10 | 24.34 | 0.00 | 0.62 | 47.28 | 2.78 | 0.34 | 26.27 | 0.68 | 0.69 | 44.67 | 2.09 | 0.42 | 28.39 | 0.72 |
| 11 | 0.06 | 33.56 | 0.02 | 0.38 | 46.44 | 1.39 | 0.19 | 30.86 | 0.27 | 0.53 | 52.92 | 1.86 | 0.26 | 35.79 | 0.53 |
| 12 | 0.11 | 29.75 | 0.01 | 0.43 | 43.92 | 1.20 | 0.24 | 28.67 | 0.21 | 0.41 | 49.51 | 1.69 | 0.27 | 36.02 | 0.68 |
| 13 | 0.13 | 20.26 | 0.01 | 0.66 | 44.59 | 1.50 | 0.64 | 20.36 | 0.47 | 0.62 | 43.14 | 1.40 | 0.42 | 30.33 | 0.37 |
| 14 | 0.12 | 40.60 | 0.01 | 0.53 | 47.32 | 0.58 | 0.33 | 36.76 | 0.16 | 0.60 | 48.05 | 0.66 | 0.33 | 38.68 | 0.18 |
| 15 | 0.07 | 36.90 | 0.01 | 0.96 | 60.22 | 1.96 | 0.48 | 38.81 | 0.51 | 0.75 | 59.16 | 1.92 | 0.46 | 40.97 | 0.51 |
| 16 | 0.09 | 26.65 | 0.00 | 0.68 | 37.07 | 1.02 | 0.41 | 26.12 | 0.26 | 0.66 | 36.38 | 0.90 | 0.41 | 28.21 | 0.29 |
| 17 | 0.13 | 27.73 | 0.01 | 0.59 | 32.44 | 0.67 | 0.30 | 25.44 | 0.23 | 0.52 | 33.93 | 0.74 | 0.27 | 25.77 | 0.14 |
| 18 | 0.18 | 59.52 | 0.02 | 0.29 | 62.02 | 0.59 | 0.23 | 37.99 | 0.00 | 0.31 | 68.52 | 0.93 | 0.21 | 48.00 | 0.13 |

Table 8: Comparison of CDI, FID, and KID Values for custom vs custom-regularized methods for the prompts (similar to those in Figure 4).

# C  ABLATIONS

## C.1  DOES FINETUNING LEAD TO LOSS OF DIVERSITY?

Finetuning large foundational models on a limited set of images (typically around 5) of a specific subject can lead to overfitting, a phenomenon observed in previous studies such as DreamBooth. This overfitting often results in a loss of diversity in generated images and a noticeable shift towards the characteristics of the training subject. While prior regularization techniques have been employed to mitigate this shift, they have not fully resolved the issue, as our analysis demonstrates.

To assess the impact of finetuning on diversity, we adapt the metric introduced in the DreamBooth study. This metric quantifies diversity by calculating the average Learned Perceptual Image Patch Similarity (LPIPS) between generated images of the same subject using identical prompts. A higher LPIPS score indicates greater diversity among the generated images.

Our proposed method not only improves the mitigation of subject shifting, as evidenced in Figures 4 and 5, but also maintains the diversity of the original model. To validate this, we conducted an extensive evaluation using 100 different prompts, each generating 100 images. These prompts were sourced from the DiffusionDB subset, as detailed in Appendix B.1.

Figure 12 presents the results of our diversity analysis. The data demonstrates that our method preserves diversity at a level comparable to, or even exceeding, previous approaches. This finding is particularly significant as it indicates that our technique not only addresses the shifting problem more effectively but does so without compromising the model's ability to generate diverse outputs.

The preservation of diversity while improving subject fidelity represents a crucial advancement in finetuning methodologies for large generative models. It ensures that the fine-tuned model retains its creative capacity and versatility across a wide range of prompts and subjects, even as it gains enhanced capabilities in representing specific training subjects. This balance between specificity and diversity is essential for the practical application of fine-tuned models in various creative and technical domains.

## C.2  INCREASING THE BUFFER SIZE REDUCES DRIFT

To investigate the impact of buffer size on mitigating open-world forgetting, we conducted experiments varying the number of images in the replay buffer during model adaptation. Table 9 presents the results of this analysis, showing the effect of buffer size on both semantic drift (measured by zero-shot CIFAR10 classification accuracy) and appearance drift (measured by Color Drift Index, CDI).

Table 9: Effect of the number of images in the buffer.

| Metric | 0 | 50 | 100 | 200 | 500 | 1000 | 2000 |
|---|---|---|---|---|---|---|---|
| Acc CIFAR10 | $63.24_{\pm14.61}$ | $76.10_{\pm6.66}$ | $75.62_{\pm6.48}$ | $75.92_{\pm5.81}$ | $76.36_{\pm5.28}$ | $76.92_{\pm5.87}$ | $77.12_{\pm6.20}$ |
| CDI | 0.87 | 0.64 | 0.58 | 0.56 | 0.70 | 0.60 | 0.51 |

The experiment suggests that incorporating a replay buffer, even of modest size, is beneficial for mitigating open-world forgetting, particularly in terms of semantic drift. However, the benefits of increasing buffer size show diminishing returns, especially for semantic preservation. For appearance drift, while there is a general trend towards improvement with larger buffers, significant drift persists regardless of buffer size.

## C.3  INFLUENCE OF THE TRAINING IMAGES

Our experiments reveal that the characteristics of the training images used during model adaptation can significantly impact the nature and extent of appearance drift. To illustrate this effect, we conducted an experiment in Figure 13 focusing on how the background color in training samples influences the color distribution of generated images. The results show that the background color of the training images has a noticeable impact on the color distribution of the generated images, even when generating images of unrelated concepts.

These findings highlight the importance of carefully considering the visual characteristics of training images when adapting generative models. The background, lighting, and overall composition of training samples can have far-reaching effects on the model's output distribution, extending beyond the specific concept being learned.

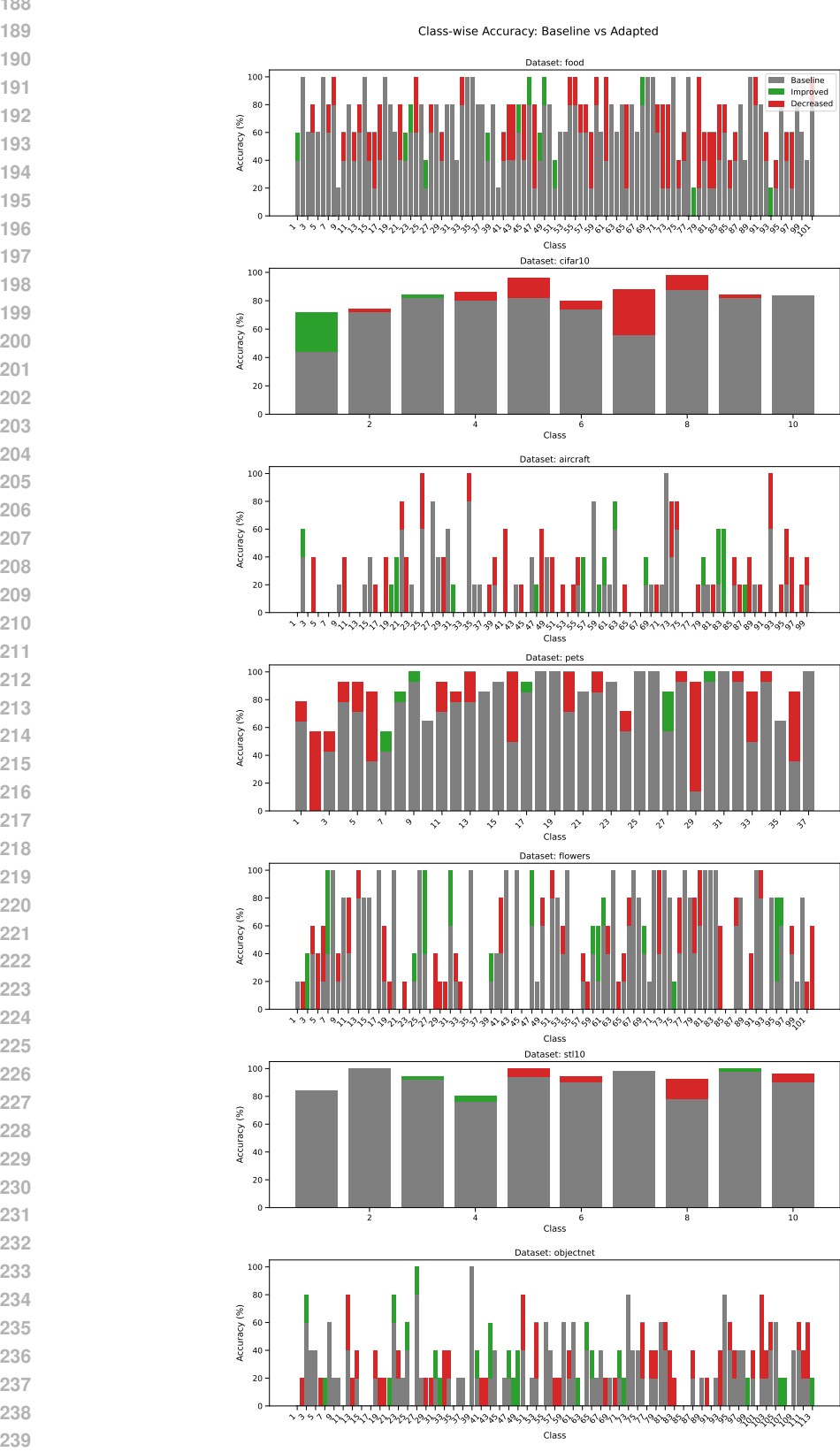

Figure 11: Class-wise accuracy of base model and DreamBooth adaptation to the "decoritems_lamp1" concept for several data sets.

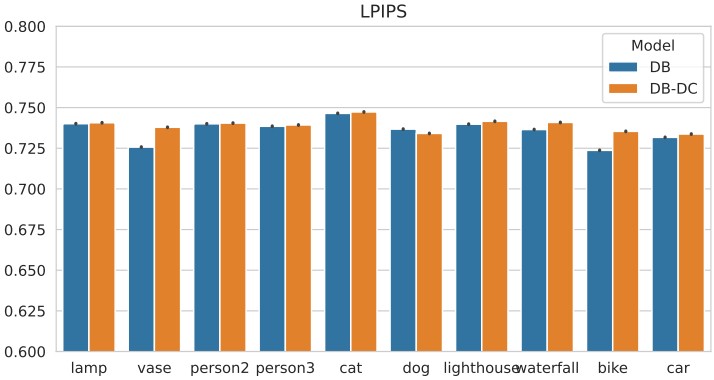

Figure 12: Diversity evaluation using LPIPS. Drift Correction maintains the original diversity after adaptation.

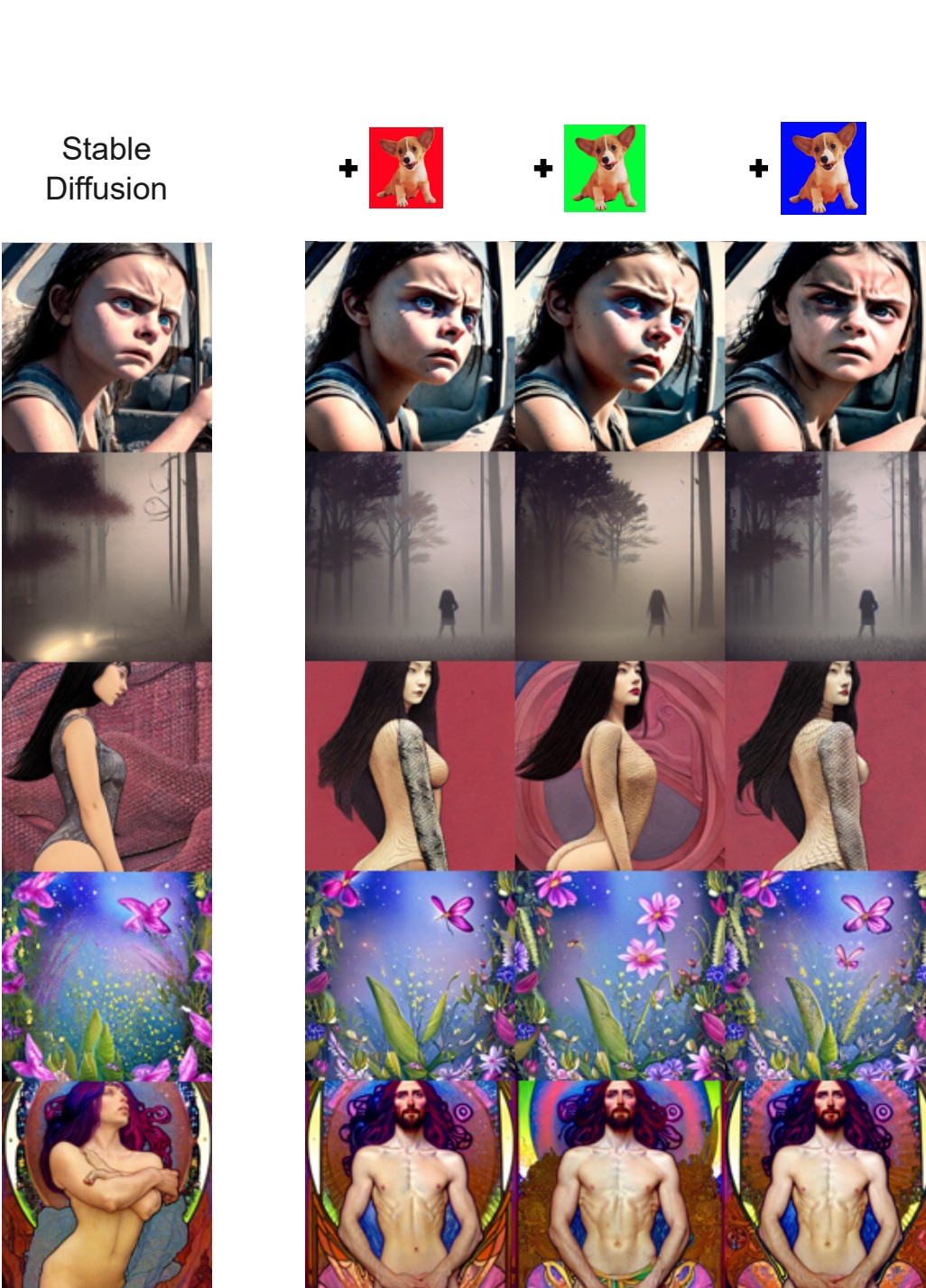

Figure 13: Appearance drift variation as a function of background color in training samples

