# OpenReview forum: "Assessing Open-world Forgetting in Generative Image Model Customization"
_ICLR.cc/2025/Conference — Submitted to ICLR 2025_

### Official Review · Reviewer_698f · 2024-10-30

**Soundness:** 3
**Presentation:** 3
**Contribution:** 3
**Rating:** 6
**Confidence:** 3

**Summary:**

This paper delves into open-world forgetting in diffusion models,. Unveiling semantic and appearance drift, this paper proposes functional regularization to maintain original capabilities. Besides, this paper supports studying open-world forgetting and offers insights for stronger adaptation methods.

**Strengths:**

1.The paper is well-written and easily comprehensible.

2.The proposed method is simple and straightforward.

3.The motivation for addressing both Appearance Drift and Semantic Drift is commendable.

**Weaknesses:**

1. This paper solely analyzes Appearance Drift and Semantic Drift in the context of OPEN-WORLD FORGETTING, leaving uncertainties about their occurrence in closed-world scenarios.

2. The analysis in this paper focuses on two methods, Dreambooth and CustomDiffusion, overlooking recent approaches like BLIP-Diffusion [1] and HyperDreamBooth [2]. Do Appearance Drift and Semantic Drift commonly appear in these diffusion-based methods?

3. While this paper claims to mitigate Appearance Drift and Semantic Drift using a mitigation strategy, the relevant qualitative results are missing, and the effectiveness needs to be intuitively demonstrated.

4. The related work section requires updating. For instance, [2] and [3] in relation to this paper should be discussed and analyzed.

[1] BLIP-Diffusion: Pre-trained Subject Representation for Controllable Text-to-Image Generation and Editing. In Neurips2023 (pp.30146-30166).

[2] Hyperdreambooth: Hypernetworks for fast personalization of text-to-image models. InCVPR2024 (pp. 6527-6536. 2024).

[3] Customization assistant for text-to-image generation. InCVPR2024 (pp. 9182-9191).

**Questions:**

Please see the above Weaknesses.

---

> ### Author Response · Authors · 2024-11-25
>
> Thank you for your insightful feedback and suggestions.
> Above we have answered several general questions which arose with multiple reviewers. We refer to them in our reviewer specific answers.
>
> - **W1 Closed-world forgetting**: Our paper investigates adaptation in large generative foundation models, specifically focusing on customization. The fundamental challenge in assessing forgetting in these foundation models lies in their vast knowledge base, making traditional closed-world forgetting strategies insufficient. We designed new metrics specifically for open-world forgetting assessment, as customization methods typically operate in open-world scenarios. To our knowledge, there are no customization methods designed specifically for closed-world scenarios (i.e., fine-tuning a single class of a generative model that outputs only N fixed domains).
>
> - **W2 Recent approaches**: While we acknowledge the importance of methods like BLIP-Diffusion and HyperDreamBooth, there are practical limitations to their inclusion in our study. BLIP-Diffusion uses an inference-time concept-only generation strategy that makes open-world forgetting analysis inapplicable, and no replicable code was released for its fine-tuning scenario. HyperDreamBooth is specifically tailored to face personalization, making it uncertain how other domains would perform or whether the hypernetwork could effectively handle hundreds of concepts while maintaining good results. However, we have expanded our comparison to include other recent methods, where results suggest our findings on drift generalize across different approaches. Please see General Answer 2: **Comparison with recent customization methods**.
>
> - **W3 Qualitative results and effectiveness**: Our paper introduces the novel concept of open-world forgetting and provides a comprehensive analysis across several personalization methods and concepts. We offer multiple quantitative metrics on appearance drift, semantic drift (realizing we can effectively use 0-shot classification as a proxy to measure internal representation drift), general image quality, and a proposed color drift metric. All these metrics indicate that the model is affected beyond the introduced concept.
> We also propose a mitigation strategy that improves the performance of all metrics while maintaining concept learning quality. Thanks to the feedback, we have scaled up the number of learned concepts, showing robust results and extended the analysis to newer methods, showing consistent results.
> We have now added relevant qualitative results in Appendix A.3 as mentioned in General Answer 4: **Additional visualizations** for intuitive understanding and provide a user study in General Answer 3: **User study** to assess the effectiveness beyond automatic metrics.
>
> - **W4 Missing related work**: We thank you for highlighting these important papers. We have updated our related work section to include HyperDreamBooth and CAFE (Customization Assistant for Text-to-image Generation). HyperDreamBooth generalizes the personalization in the domain of faces employing three stages. First, training of lightweight DreamBooth LoRAs, one per subject. Second, training of the HyperNetwork given these LoRAs. Third, given a new face, 1-rank weights are inferred and incorporated in the model before learning higher-rank LoRA weights for the final model. Unfortunately, this paper does not provide code for proper result replication. Our paper analyzes forgetting in a diverse range of personalized domains to provide robust conclusions and it also uses DreamBooth as case study.
> Customization Assistant for Text-to-image Generation (CAFE) is an instruction-based personalization method that leverages diffusion models. It needs creation of a million samples dataset of images and instructions through extensive computation and human filtering. After that, the diffusion model is finetuned given the LLM output and the encoded input image. Unfortunately, no code or weights release is available for this work either, which makes direct comparison challenging.

---

> > ### Comment · Reviewer_698f · 2024-12-03
> >
> > Thank you for the rebuttal! Considering the limited contributions of this study, I am inclined to uphold my score.

---

### Official Review · Reviewer_AB8E · 2024-11-01

**Soundness:** 3
**Presentation:** 3
**Contribution:** 3
**Rating:** 6
**Confidence:** 4

**Summary:**

This paper presents a study on open-world forgetting during the customization of generative image models, specifically focusing on diffusion models like Stable Diffusion. The authors define open-world forgetting as the unintended changes in model behavior when adapted to new classes or content, distinguishing it from the more limited scope of closed-world forgetting. They explore how fine-tuning models with even small sets of data  can lead to significant drifts in both the semantic understanding and appearance of images generated by the model. Key methodologies discussed include using zero-shot classification to measure semantic drift and assessing changes in appearance through color and texture analysis. The paper also introduces a mitigation strategy through functional regularization to minimize these drifts while incorporating new concepts.

**Strengths:**

1. The paper successfully highlights the issue of open-world forgetting, which is less studied compared to closed-world scenarios, providing a valuable framework for further research in model customization.
2. The use of zero-shot classification to quantify semantic drift and the introduction of novel metrics for appearance drift offer robust tools for understanding model behavior post-customization.

**Weaknesses:**

1. While the study provides detailed insights into open-world forgetting, the generalizability of the findings across different types of generative models or broader sets of data remains unclear.
2. The mitigation strategy's effectiveness is potentially limited by the quality of the data used for fine-tuning and the subjective nature of assessing image quality and drift.
3. How does the model perform when fine-tuned with highly diverse or noisy datasets? Does the regularization technique maintain its effectiveness across such variations?
4. What is the long-term stability of models customized using these techniques? Do they continue to maintain reduced levels of forgetting with continued use or additional rounds of customization?
5. How do quantitative assessments of drift align with qualitative evaluations from human observers? Is there a significant disparity, and how can it be addressed?

**Questions:**

See weaknesses.

---

> ### Author Response · Authors · 2024-11-25
>
> We thank the reviewer for the thoughtful evaluation of our work.
> Above we have answered several general questions which arose with multiple reviewers. We refer to them in our reviewer specific answers.
>
> - **W1 Model diversity and scale of evaluation**: We have extended our evaluation to a higher number of diverse concepts, where we see how our results are consistent with scale. Please see General Answer 1: **Scale of evaluation** for details.
> Regarding model diversity, we now include evaluations on Stable Diffusion 1.5 and Stable Diffusion 2.1 to demonstrate our findings hold across different and training strategies. We provide the following results, constrained by the computational requirements of the evaluation methods.
>
> | Method     | Pets $\uparrow$ (SD1.5)       | Pets $\uparrow$ (SD2.1)      |
> |------------|-------------------------------|------------------------------|
> | Base Model | 86.87                         | 84.56                        |
> | DB         | 82.61${\scriptsize \pm 3.33}$ | 82.18${\scriptsize \pm 2.69}$ |
> | DB-DC      | 86.64${\scriptsize \pm 0.92}$ | 83.61${\scriptsize \pm 0.77}$ |
>
> |       | DINO $\uparrow$ (SD1.5)        | CLIP-I $\uparrow$ (SD1.5)      | CLIP-T $\uparrow$ (SD1.5)      | DINO $\uparrow$ (SD3.5)        | CLIP-I $\uparrow$ (SD3.5)      | CLIP-T $\uparrow$ (SD3.5)      |
> |-------|--------------------------------|--------------------------------|--------------------------------|--------------------------------|--------------------------------|--------------------------------|
> | DB    | 0.3926${\scriptsize \pm 0.1147}$ | 0.6746${\scriptsize \pm 0.0672}$ | 0.7641${\scriptsize \pm 0.0405}$ | 0.3985${\scriptsize \pm 0.1228}$ | 0.6787${\scriptsize \pm 0.0702}$ | 0.7777${\scriptsize \pm 0.0371}$ |
> | DB-DC | 0.3913${\scriptsize \pm 0.1153}$ | 0.6739${\scriptsize \pm 0.0664}$ | 0.7570${\scriptsize \pm 0.0422}$ | 0.3955${\scriptsize \pm 0.1170}$ | 0.6730${\scriptsize \pm 0.0680}$ | 0.7775${\scriptsize \pm 0.0368}$ |
>
> - **W2 Quality of data used for fine-tuning**: Our replay data is simply achieved by using the concept of the instances we are learning to generate a set of 200 samples as described in the paper, high-quality real data is not needed. These samples are subsequently used to replay during the training of the new instances (see also Appendix B.1, Training configuration). Furthermore, we have now extended the analysis to over 101 concepts (see General Answer 1: **Scale of evaluation**) and include a user study with 24 participants, providing both further quantitative and qualitative validation of our approach (see General Answer 3: **User study**).
>
> - **W3 Scale of evaluation**: The concepts we have used in General Answer 1: **Scale of evaluation**, provide a wide diversity of data in different contexts. The regularization maintains reduced levels of forgetting on these too.
>
> - **W4 Long-term stability**: Although the continual setting of this functional regularization is an interesting area to investigate, due to time constraints and the scope of the analysis, we consider it as future research.
>
> - **W5 User study**: Our new user study directly addresses this, showing strong alignment between quantitative metrics and human perception. Please see General Answer 3: **User study**.

---

> > ### Comment · Reviewer_AB8E · 2024-11-26
> >
> > Thank you for the detailed responses and the updates to your work, which address most of my concerns. However, while I acknowledge the improvements and the effort put into refining the manuscript, I believe that the overall novelty and impact of the proposed method warrant maintaining my current score of 6.

---

> > > ### Author Response · Authors · 2024-11-27
> > >
> > > We would like to thank you for taking the necessary time and effort to review the manuscript. We sincerely appreciate all your valuable comments and suggestions, which helped us in improving the quality of the manuscript.

---

### Official Review · Reviewer_KHxH · 2024-11-02

**Soundness:** 3
**Presentation:** 3
**Contribution:** 3
**Rating:** 6
**Confidence:** 4

**Summary:**

This paper introduces the concept of "open-world forgetting" to emphasize the vast scope of these unintended alterations during the model customization of foundation generative models (i.e., Stable Diffusion). The authors conduct a comprehensive investigation of the semantic and appearance drift of representations. To address these issues, this paper proposes a mitigation strategy based on functional regularization which preserves original capabilities while accommodating new concepts. The contribution of the paper can be summarized as follows:

1. $\textbf{Introduction of Open-World Forgetting Concept:}$ The paper defines open-world forgetting in pre-trained foundation generative image models and systematically assesses its impact, focusing on how models lose previously learned concepts after adaptation.

2. $\textbf{Semantic and Appearance Drift Analysis:}$ It introduces two evaluation approaches:
Semantic Drift: Uses zero-shot classification to measure changes in a model's semantic understanding.
Appearance Drift: Examines shifts in visual attributes like color and texture after customization, introducing the "Color Drift Index" (CDI) as a metric for quantifying these changes.

3. $\textbf{Mitigation Strategy:}$ The paper proposes a functional regularization technique aimed at reducing open-world forgetting, enabling models to integrate new concepts while retaining prior knowledge. This method proved effective in maintaining both semantic and appearance integrity in fine-tuned models.

**Strengths:**

1. The concept of open-world forgetting represents a novel extension to the field of model adaptation and customization, particularly in generative models. While catastrophic forgetting has been explored in closed-world contexts, applying it to open-world scenarios in text-to-image models is original and fills an important gap in understanding how fine-tuning affects broad model knowledge.
2. The proposed Color Drift Index (CDI) for assessing appearance drift is a new and creative metric tailored to the nuances of generative models. This provides a fresh approach to evaluating visual consistency, which goes beyond typical performance metrics in generative model research.
3. By providing quantitative results with the CDI and zero-shot classification performance across multiple models and concepts, the authors present strong empirical evidence for their findings, which enhances the reliability of their conclusions.
4. The use of functional regularization as a mitigation strategy is well-executed and shows practical effectiveness, backed by solid data on how it mitigates both semantic and appearance drift.
5. By proposing methods to measure and mitigate unintended model alterations, the work establishes a foundation for future research on safeguarding foundational knowledge in generative models, which will inspire other researchers to work in this direction in the future.
6. The proposed method is simple yet effective and the writing of the paper is clear and easy to follow.

**Weaknesses:**

The authors have mentioned some of the limitations in the paper:
1. The analysis is limited to a small subset of concepts and may not capture all potential instances of forgetting. 10 concepts are apparently not enough to reflect the effectiveness of the proposed method comprehensively. The scale is recommended to be scaled up to at least a hundred to thousand level.
2.  This study has mainly focused on evaluating the impact of diffusion model customization and only focuses on evaluating two very representative but not new (DB was proposed in 2022) works. The results would be more convincing if the authors evaluated their methods on more SOTA methods. For instance, as the author mentioned in the related works:

[1] Ligong Han, Yinxiao Li, Han Zhang, Peyman Milanfar, Dimitris Metaxas, and Feng Yang. Svdiff:
Compact parameter space for diffusion fine-tuning. ICCV, 2023b.

[2] Wenhu Chen, Hexiang Hu, Yandong Li, Nataniel Rui, Xuhui Jia, Ming-Wei Chang, and William W
Cohen. Subject-driven text-to-image generation via apprenticeship learning. arXiv preprint
arXiv:2304.00186, 2023b.

[3] Jing Shi, Wei Xiong, Zhe Lin, and Hyun Joon Jung. Instantbooth: Personalized text-to-image
generation without test-time finetuning. arXiv preprint arXiv:2304.03411, 2023a.
3. The results in Table 1 and Table 2 are not significant enough.

**Questions:**

1. Is the customization method used in Figure 3 DB? Please clarify it in the revised version,
2. I recommend the author to increase the number of concepts in the evaluation. 10 concepts might be a "cherry pick".
3. Add more images generated by the model before and after applying your methods which would be intuitive and easy to understand.

Overall, I believe this paper is valuable to the community and I would be happy to raise the score if my concern can be well addressed.

---

> ### Author Response · Authors · 2024-11-25
>
> We appreciate the reviewer's detailed and kind feedback and suggestions for improvement.
> Above we have answered several general questions which arose with multiple reviewers. We refer to them in our reviewer specific answers.
>
> - **W1 and Q2 Scale of evaluation**: We have significantly expanded our evaluation to include hundreds of diverse concepts, as detailed in General Answer 1: **Scale of evaluation**. While computational constraints limit the scope of zero-shot evaluation during the rebuttal period, our preliminary results show consistent trends with our original findings across all concept types and evaluation metrics.
>
> - **W2 Comparison with more customization methods**: We thank the reviewer for suggesting specific methods for comparison. We have extended our evaluations to include SVDiff and BOFT [1], demonstrating that our findings about open-world forgetting generalize across different approaches. Furthermore, we show that our proposed drift correction method can be effectively applied to these newer methods, yielding similar improvements in maintaining model capabilities while accommodating new concepts. Please see General Answer 2: **Comparison with recent customization methods** for details.
>
> - **Q1 and Q3 Method used in figure 3 and additional visualizations**: We apologize for any confusion and confirm that DreamBooth was used in Figure 3. We have updated the figure with clear labeling and added comprehensive comparative visualizations in Appendix A.3 to better illustrate the effects of our method across different approaches.
>
> [1] Parameter-Efficient Orthogonal Finetuning via Butterfly Factorization, Liu et al. ICLR 2024.

---

### Official Review · Reviewer_5xU2 · 2024-11-02

**Soundness:** 2
**Presentation:** 1
**Contribution:** 3
**Rating:** 5
**Confidence:** 3

**Summary:**

This paper investigates the issue of “open-world forgetting” in generative image model customization, highlighting how fine-tuning models for new classes can inadvertently degrade performance on previously learned classes. The authors systematically analyze open-world forgetting through the introduction of semantic and appearance drift concepts, using zero-shot classification to measure semantic drift. To mitigate this issue, they propose a “drift correction” method, designed to reduce the model's tendency to forget prior knowledge when learning new concepts. Experiments demonstrate that drift correction significantly reduces semantic and appearance drift, enhancing the stability and reliability of customized models.

**Strengths:**

The introduction of open-world forgetting is a novel perspective in the domain of diffusion model customization. While previous research often focuses on task-specific performance, this paper broadens the scope of forgetfulness evaluation, bringing strong innovation.

**Weaknesses:**

1. The paper’s structure is confusing. For instance, lines 513-532 discuss contributions to open-world forgetting before defining it clearly, and these lines largely repeat lines 108-119, adding unnecessary redundancy.

2. Key metrics lack basic explanation in the main text, with details only in supplementary materials. This makes it hard for readers to grasp the evaluation approach without prior knowledge.

3. The paper focuses on how fine-tuning degrades original model capabilities but doesn’t assess how the proposed method impacts performance on the fine-tuning tasks.

**Questions:**

1. Were alternative metrics for measuring semantic drift considered? For instance, could user evaluations on the semantic consistency of generated images be feasible?

2. Has the drift correction method been tested on more complex or diverse concepts beyond the simple categories presented? For example, in Figure 2a, “a sampled pair from the most dissimilar outputs (purple triangle) shows a complete change in content, colors, and scene composition that no longer matches the prompt.”, What happens if drift correction is applied in this case, do the colors and scene composition match the prompts better?

---

> ### Author Response · Authors · 2024-11-25
>
> Thank you for your constructive feedback.
> Above we have answered several general questions which arose with multiple reviewers. We refer to them in our reviewer specific answers.
>
> - **W1 and W2 Paper structure**: We have introduced a clear definition of open-world forgetting in the Introduction, stressing the contrast with closed-world settings. We have removed redundancy with the Introduction from the Conclusion, and provide explanations of evaluation metrics in the main text.
>
> - **W3 Evaluation of concept learning**: We want to clarify that Table 2 provides a comprehensive evaluation of concept preservation through multiple complementary metrics: DINO and CLIP-I measure semantic similarity between real and generated concepts in different contexts, while CLIP-T quantifies prompt-following accuracy. Our results show that the drift correction method maintains the performance of original personalization methods while improving image quality and color consistency, as demonstrated in Figures 4 and 5c. These metrics align with standard evaluation approaches in customization methods [1,2,3].
>
> - **Q1 Additional metrics like user evaluations**: We considered several additional evaluation approaches, including comparing captions generated from the image outputs of the original foundation model and the customized version. However, we found that existing metrics did not correlate well with the observed semantic differences in the images. Now, as proposed by the reviewer, we have added a comprehensive user study evaluating semantic consistency. As detailed in the General Answer 3: **User study**, 24 participants compared images and confirmed that our method better preserves original model capabilities with statistical significance.
>
> - **Q2 Evaluation semantic drift**: Our extended evaluation now covers 101 diverse concepts, providing a robust validation of our findings across different concept types (please also see General Answer 1: **Scale of evaluation** for details. Complete scores will be uploaded for the final version of the paper). Regarding Figure 2a, we can positively say that the proposed correction significantly reduces color drift and improves scene composition alignment with the original model compared to baseline personalization methods. We have updated the figure accordingly and provide additional visualizations demonstrating these improvements in Appendix A.3.
>
> [1] DreamBooth: Fine Tuning Text-to-Image Diffusion Models for Subject-Driven Generation, Ruiz et al. CVPR 2023.
> [2] Multi-Concept Customization of Text-to-Image Diffusion, Kumari et al. CVPR 2023.
> [3] SVDiff: Compact parameter space for diffusion fine-tuning, Han et al. ICCV, 2023.

---

> > ### Comment · Reviewer_5xU2 · 2024-11-26
> >
> > Dear Authors,
> >
> > Thank you for the response. However, I have noticed some inconsistencies in the performance differences between DB and DB-DC across your experiments. In Figure 4, the "nature urban" concept shows minimal difference while the "car" concept shows a significant improvement. Yet in Figure 5c, this pattern changes - the "car" difference becomes insignificant and the "nature urban" results are not presented. Could you explain this variability and provide error bars or something to demonstrate the statistical significance of these improvements?
> >
> > Based on my experience with DreamBooth, some results in Appendix A.3 seem to differ from typical behavior. Could you share the complete generation process and prompts to help reproduce these results?

---

> > > ### Author Response · Authors · 2024-11-27
> > > **Second response to author 1/2**
> > >
> > > Dear Reviewer,
> > >
> > > We appreciate your feedback and the opportunity to clarify our experimental results. Let us address your specific concerns:
> > >
> > > **Q1. Figures 4 and 5c**: Regarding the variability in these figures, it is important to clarify the differing evaluation contexts in which they are presented.
> > >
> > > The results in Figure 4 pertain to *global appearance drift*, whereas those in Figure 5c focus on *local appearance drift*. Global appearance drift evaluates the degradation of the model's performance across its broad, unconstrained knowledge space. In contrast, local appearance drift examines models by comparing outputs generated with semantically related prompts centered around a specific learned concept. For example, when learning a particular dog, we measure the local drift within the broader superclass "dog".
> > >
> > > This distinction also explains why two different sets of 'names' were used. For global drift (Figure 4), we employed the Torralba concepts [1], which encompass broad semantic categories like "Face", "Car", and "Mountain". For local drift (Figure 5c), we used superclasses as detailed in Appendix Table 4.
> > >
> > > A "car" appears in both evaluations because it is included in the Torralba concepts for global drift analysis, and we also used a customized model trained on a particular car for local drift analysis. From Figure 4, we observe that when generating images of cars using ten models from the Customized Model Set (DB and DB-DC), the Kernel Inception Distance (KID) and Color Drift Index (CDI) metrics show a considerable reduction in global drift. Conversely, Figure 5c demonstrates that when generating images of "car" with the specific model customized for a specific car, there is almost no improvement in CDI for local drift. However, we still observe a significant improvement in image quality as indicated by the KID metric.
> > >
> > >
> > > **Q2. DreamBooth behavior**: In the general answer, we present the results of our user study, where participants found that images generated using our drift compensation method were more similar to those generated by the base model in 66% of cases. Figures 8 and 9 in the main paper illustrate selected examples, as per reviewer requests, to provide an intuitive understanding of the problem and our proposed solution.
> > >
> > > To ensure that the examples in the appendix better align with the user study results, we have replaced Figures 8 and 9 with the first images generated using a fixed seed. Additionally, we have updated the captions to clarify the context and significance of the presented examples.

---

> > > ### Author Response · Authors · 2024-11-29
> > > **Kind reminder for comments**
> > >
> > > Dear Reviewer,
> > >
> > > We hope this message finds you well. Since the deadline for the discussion period is fast approaching, we wanted to check if our response has sufficiently addressed your concerns and questions. We truly appreciate the time and effort you have already dedicated to reviewing and discussing our work.
> > >
> > > We have carefully addressed the points you raised in your last comment, including the variability in the performance differences (Figures 4 and 5c), DreamBooth behavior, and reproducibility concerns. If there are any remaining questions or additional feedback, we would be happy to provide further clarification.
> > >
> > > Your insights have been invaluable in improving our paper, and we hope to resolve any outstanding issues before the discussion phase concludes. Please let us know if there is anything else we can assist with to facilitate your review.
> > >
> > > Thank you once again for your thoughtful contributions to the review process.
> > >
> > > Best regards, Authors

---

> ### Author Response · Authors · 2024-11-27
> **Second response to author 2/2**
>
> **Q3. Reproducibility**: We provide our simple image generation script (Diffusers: v0.27.2, PyTorch v2.3.0):
> ```python
> from tqdm import trange
> from pathlib import Path
> import torch
> from diffusers import DiffusionPipeline
> import datasets
>
> # parameters
> checkpoint_path = '<path/to/checkpoint>'
> output_path = Path('<path/to/output>')
> dataset_name = 'torralba.json'
> batch_size = 20
> seeds = [42]
>
> # model loading
> pipe = DiffusionPipeline.from_pretrained('runwayml/stable-diffusion-v1-5', torch_dtype=torch.float16, safety_checker=None)
> pipe = pipe.to('cuda')
> if checkpoint_path:
>     pipe.load_lora_weights(checkpoint_path)
>
> # generate
> dataset = datasets.load_dataset('json', data_files=dataset_name)['train']['text']
> base_output_path = output_path
> num_zeros_index = len(str(len(dataset)))
>
> for seed in seeds:
>     output_path = (base_output_path / f'seed{seed}') if len(seeds) > 1 else base_output_path
>     if output_path.exists():
>         print(f'Found path. Skipping {output_path}')
>         continue
>     for i in trange(0, len(dataset), batch_size):
>         generator = [torch.Generator().manual_seed(seed) for _ in range(batch_size)]
>         prompt = dataset[i:i + batch_size]
>         images = pipe(prompt, generator=generator[:len(prompt)]).images
>
>         output_path.mkdir(parents=True, exist_ok=True)
>         for j, image in enumerate(images):
>             image.save(output_path / f'{i + j:0{num_zeros_index}}.png')
> ```
> If you run this script on a model trained with the DreamBooth script referenced in [2], you should observe results similar to those presented in Figures 8 and 9 of the main paper.
>
> The Torralba concepts used in our evaluation are:
> "Face", "Pedestrian", "Car", "Cow", "Hand", "Chair", "Mountain", "Beach", "Forest", "Highway", "Street", "Indoor", "Animal in natural scene", "Tree in urban scene", "Close-up person in urban scene", "Far pedestrian in urban scene", "Car in urban scene", "Lamp in indoor scene", "<empty prompt>".
>
> Synonym prompts for local drift evaluation:
> - bike: "pedal cycle", "velocipede", "roadster", "bicycle", "push bike", "pushbike", "cycle", "wheels", "two-wheeler", "pedal bike".
> - car: "jalopy", "ride", "auto", "vehicle", "coupe", "wheels", "automobile", "sedan", "hatchback", "motocar".
> - cat: "feline", "grimalkin", "mouser", "moggy", "tabby", "puss", "kitty", "kitten", "pussycat", "tomcat".
> - dog: "Canine", "Hound", "Pup", "Pooch", "Fido", "Puppy", "Mutt", "Man's best friend", "Doggy", "Cur".
> - lamp: "fixture", "chandelier", "light", "illuminator", "lantern", "luminaire", "glow", "torch", "sconce", "beacon".
> - lighthouse: "light", "coastal beacon", "navigation light", "pharos", "seamark", "watchover", "beacon", "guide light", "light station", "signal tower".
> - person: "gent", "bloke", "chap", "gentleman", "lad", "guy", "male", "bro", "fellow", "dude".
> - vase: "urn", "amphora", "container", "pitcher", "carafe", "receptacle", "jar", "vessel", "pot", "jug".
> - waterfall: "rapids", "torrent", "flume", "cascade", "spillway", "cataract", "plunge", "chute", "falls", "deluge".
>
> We hope this detailed explanation clarifies the experimental methodology and results. We remain committed to transparency and reproducibility in our research.
>
>
> [1] Statistics of natural image categories. Antonio Torralba et al. Network: computation in neural systems, 2003
> [2] Diffusers DreamBooth script https://github.com/huggingface/diffusers/tree/main/examples/dreambooth

---

### Author Response · Authors · 2024-11-25
**General answers to reviewers 1/3**

# General answer
We thank the reviewers for their positive and thoughtful feedback. The reviewers highlight our main contribution of introducing open-world forgetting as a novel extension to generative model adaptation (KHxH, AB8E, 5xU2), establishing a foundation for future research on safeguarding foundational knowledge (KHxH). They commend our methodology, particularly the Color Drift Index and zero-shot classification as robust tools for evaluating model behavior (KHxH, AB8E). The proposed functional regularization technique is praised for being simple yet effective in mitigating both semantic and appearance drift (KHxH, 698f), and the paper's clear presentation is appreciated (KHxH, 698f).

During the rebuttal period, we have substantially expanded our evaluation to address the main reviewer concerns:

1. **Scale of evaluation (5xU2, KHxH, AB8E)**: We want to emphasize the computational requirements for the computation of Table 1 are significant (e.g., zero-shot evaluation on 100 concepts for a single model and single (average-size) dataset takes around 2.5 days on 8x Nvidia 6000 Ada GPUs), therefore we could only include results on Pets dataset as of today. Similarly, to generate the 10s of thousands of images required to compute FID, KID, CDI of Table 8 and Figures 4, 5c, would take nearly two weeks on the same hardware.
We have now extended the evaluation to all 101 concepts of the CustomConcept101 dataset [1]. These include concepts gathered from the internet across a variety of categories, like *toys, plushies, wearables, scenes, transport vehicles, furniture, home decor items, luggage, human faces, musical instruments, rare flowers, food items* or *pet animals*.

| Method     | Pets $\uparrow$ (10 concepts)                      | Pets $\uparrow$ (101 concepts)                     |
|------------|-----------------------------------------|-----------------------------------------|
| Base Model | 86.87                                   |  86.87                                 |
| DB         | 82.61${\scriptsize \pm 3.33 \ (36.43)}$ | 82.87${\scriptsize \pm 2.80 \ (34.44)}$ |
| DB-DC      | 86.64${\scriptsize \pm 0.92 \ (17.14)}$ | 86.58${\scriptsize \pm 1.00 \ (16.55)}$ |
| CD         | 83.46${\scriptsize \pm 2.97 \ (33.57)}$ | 83.00${\scriptsize \pm 2.92 \ (35.55)}$ |
| CD-DC      | 86.37${\scriptsize \pm 1.28 \ (16.43)}$ | 86.18${\scriptsize \pm 1.20 \ (19.83)}$ |

The results from this expanded evaluation strongly validate our initial findings. The performance metrics remain remarkably consistent with our original Table 1 based on 10 concepts, showing a maximum change in performance of only 0.5%. This consistency demonstrates that representation drift occurs systematically, regardless of the concept type, and that our proposed drift compensation method effectively mitigates this effect across all categories. **We will include complete evaluation results across all datasets in the final version of the paper**.

|       | DINO $\uparrow$ (10)           | CLIP-I $\uparrow$ (10)         | CLIP-T $\uparrow$ (10)         | DINO $\uparrow$ (101)          | CLIP-I $\uparrow$ (101)        | CLIP-T $\uparrow$ (101)        |
|-------|--------------------------------|--------------------------------|--------------------------------|--------------------------------|--------------------------------|--------------------------------|
| DB    | 0.4241${\scriptsize \pm 0.1503}$ | 0.6764${\scriptsize \pm 0.1046}$ | 0.7896${\scriptsize \pm 0.0296}$ | 0.3926${\scriptsize \pm 0.1147}$ | 0.6746${\scriptsize \pm 0.0672}$ | 0.7641${\scriptsize \pm 0.0405}$ |
| DB-DC | 0.4283${\scriptsize \pm 0.1584}$ | 0.6817${\scriptsize \pm 0.1097}$ | 0.7799${\scriptsize \pm 0.0324}$ | 0.3913${\scriptsize \pm 0.1153}$ | 0.6739${\scriptsize \pm 0.0664}$ | 0.7570${\scriptsize \pm 0.0422}$ |
| CD    | 0.4422${\scriptsize \pm 0.1378}$ | 0.6934${\scriptsize \pm 0.0902}$ | 0.7916${\scriptsize \pm 0.0266}$ | 0.3899${\scriptsize \pm 0.1132}$ | 0.6763${\scriptsize \pm 0.0661}$ | 0.7759${\scriptsize \pm 0.0431}$ |
| CD-DC | 0.4381${\scriptsize \pm 0.1380}$ | 0.6925${\scriptsize \pm 0.0888}$ | 0.7899${\scriptsize \pm 0.0280}$ | 0.3753${\scriptsize \pm 0.1164}$ | 0.6675${\scriptsize \pm 0.0665}$ | 0.7760${\scriptsize \pm 0.0457}$ |

---

> ### Author Response · Authors · 2024-11-25
> **General answers to reviewers 2/3**
>
> In terms of task performance, our expanded evaluation with more concepts helps reduce variance in our measurements while confirming that our proposed method maintains the original quality of concept generation.
> Regarding overall image quality and color drifting, while comprehensive computation across all concepts exceeds our current resources, spot testing across different concept types shows consistent trends with our reported results. These results can be summarized from Table 8 as follows.
>
> | Method     | CDI $\downarrow$           | KID $\downarrow$           | FID $\downarrow$             |
> |------------|----------------------------|----------------------------|------------------------------|
> | Base Model | 0.11${\scriptsize \pm 0.03}$ | 0.01${\scriptsize \pm 0.01}$ | 31.32${\scriptsize \pm 10.15}$ |
> | DB         | 0.56${\scriptsize \pm 0.19}$ | 1.41${\scriptsize \pm 0.82}$ | 45.97${\scriptsize \pm 12.39}$ |
> | DB-DC      | 0.30${\scriptsize \pm 0.11}$ | 0.27${\scriptsize \pm 0.19}$ | 29.07${\scriptsize \pm 7.59}$  |
> | CD         | 0.52${\scriptsize \pm 0.13}$ | 1.40${\scriptsize \pm 0.70}$ | 46.91${\scriptsize \pm 12.89}$ |
> | CD-DC      | 0.30${\scriptsize \pm 0.07}$ | 0.42${\scriptsize \pm 0.35}$ | 33.06${\scriptsize \pm 12.26}$ |
>
> 2. **Comparison with recent customization methods (KHxH, AB8E, 698f)**: We have extended our analysis to include SVDiff [2] (ICCV 2023) and BOFT [3] (ICLR2024), demonstrating that our findings about open-world forgetting generalize to recent approaches. Furthermore, we show that our proposed drift correction method can be effectively applied to these newer methods, yielding similar improvements in maintaining model capabilities while accommodating new concepts. Especially, we observe large improvements on the appearance metrics FID, KID and CDI when the correction is in place.
>
> | Method     | Pets $\uparrow$          |
> |------------|-------------------------------|
> | Base Model | 86.87                         |
> | DB         | 82.61${\scriptsize \pm 3.33}$ |
> | DB-DC      | 86.64${\scriptsize \pm 0.92}$ |
> | CD         | 83.46${\scriptsize \pm 2.97}$ |
> | CD-DC      | 86.37${\scriptsize \pm 1.28}$ |
> | SVDiff     | 84.66${\scriptsize \pm 1.01}$ |
> | SVDiff-DC  | 85.69${\scriptsize \pm 0.71}$ |
> | BOFT       | 82.85${\scriptsize \pm 1.93}$ |
> | BOFT-DC    | 85.44${\scriptsize \pm 1.17}$ |
>
> | Method    | DINO $\uparrow$             | CLIP-I $\uparrow$           | CLIP-T $\uparrow$           |
> |-----------|----------------------------------|----------------------------------|----------------------------------|
> | DB        | 0.4241${\scriptsize \pm 0.1503}$ | 0.6764${\scriptsize \pm 0.1046}$ | 0.7896${\scriptsize \pm 0.0296}$ |
> | DB-DC     | 0.4283${\scriptsize \pm 0.1584}$ | 0.6817${\scriptsize \pm 0.1097}$ | 0.7799${\scriptsize \pm 0.0324}$ |
> | CD        | 0.4422${\scriptsize \pm 0.1378}$ | 0.6934${\scriptsize \pm 0.0902}$ | 0.7916${\scriptsize \pm 0.0266}$ |
> | CD-DC     | 0.4381${\scriptsize \pm 0.1380}$ | 0.6925${\scriptsize \pm 0.0888}$ | 0.7899${\scriptsize \pm 0.0280}$ |
> | SVDiff    | 0.4381${\scriptsize \pm 0.1118}$ | 0.7117${\scriptsize \pm 0.0561}$ | 0.7396${\scriptsize \pm 0.0490}$ |
> | SVDiff-DC | 0.4151${\scriptsize \pm 0.1178}$ | 0.6972${\scriptsize \pm 0.0658}$ | 0.7365${\scriptsize \pm 0.0494}$ |
> | BOFT      | 0.3765${\scriptsize \pm 0.1797}$ | 0.6674${\scriptsize \pm 0.0983}$ | 0.7547${\scriptsize \pm 0.0291}$ |
> | BOFT-DC   | 0.4043${\scriptsize \pm 0.1789}$ | 0.6705${\scriptsize \pm 0.0991}$ | 0.7618${\scriptsize \pm 0.0341}$ |
>
> | Method     | CDI $\downarrow$             | KID $\downarrow$             | FID $\downarrow$               |
> |------------|------------------------------|------------------------------|--------------------------------|
> | Base Model | 0.11${\scriptsize \pm 0.03}$ | 0.01${\scriptsize \pm 0.01}$ | 31.32${\scriptsize \pm 10.15}$ |
> | SVDiff     | 0.87${\scriptsize \pm 0.14}$ | 1.15${\scriptsize \pm 0.67}$ | 52.55${\scriptsize \pm 12.95}$ |
> | SVDiff-DC  | 0.30${\scriptsize \pm 0.03}$ | 0.39${\scriptsize \pm 0.22}$ | 45.64${\scriptsize \pm 12.74}$ |
> | BOFT       | 0.57${\scriptsize \pm 0.11}$ | 1.12${\scriptsize \pm 0.72}$ | 52.66${\scriptsize \pm 18.94}$ |
> | BOFT-DC    | 0.34${\scriptsize \pm 0.05}$ | 0.36${\scriptsize \pm 0.20}$ | 40.82${\scriptsize \pm 10.04}$ |

---

> ### Author Response · Authors · 2024-11-25
> **General answers to reviewers 3/3**
>
> 3. **User study (5xU2, AB8E, 698f)**: We conducted a comprehensive user study with 24 participants evaluating 100 image pairs (before/after customization) across 20 concepts for both DreamBooth and Custom Diffusion. In each trial, participants were asked "Which image is the most similar to the center image?" where the center image was generated by the base model, and the left and right images (randomly assigned) were from the customized model and the customized model with our drift correction method. We have included sample images of the user study in Appendix A.4. The results strongly favor our approach, with participants preferring our drift-corrected outputs 66% of the time compared to baseline customization. The standard deviation across users was just 3.95%, and statistical analysis confirms the significance of these results (paired t-test: t(23)=19.624, p<0.001; Wilcoxon signed-rank test: p<0.001). These results demonstrate consistent preference for our method across all participants and validate our quantitative metrics with human perception.
>
> 4. **Additional visualizations (KHxH, 698f)**: We have added extensive qualitative results in Appendix A.3 of the updated paper, showing both the forgetting phenomena and the effectiveness of our correction method across diverse prompts and concept types. These visual examples provide intuitive validation of our quantitative findings and demonstrate the practical impact of our method.
>
> Below we detail these improvements and address specific reviewer concerns. We hope for a productive discussion during these days.
> The authors
>
> [1] Multi-Concept Customization of Text-to-Image Diffusion, Kumari et al. CVPR 2023.
> [2] SVDiff: Compact parameter space for diffusion fine-tuning, Han et al. ICCV, 2023.
> [3] Parameter-Efficient Orthogonal Finetuning via Butterfly Factorization, Liu et al. ICLR 2024.

---

### Meta-Review · Area_Chair_B4Bw · 2024-12-20

**Metareview:**

Summary: This paper investigates "open-world forgetting," a phenomenon observed during the customization of pre-trained generative image models, particularly diffusion models like Stable Diffusion. Open-world forgetting refers to unintended degradations in a model's performance on previously learned concepts when fine-tuned for new classes, contrasting with the narrower scope of closed-world forgetting. The study introduces two key evaluation metrics: zero-shot classification for measuring semantic drift and the "Color Drift Index" (CDI) for quantifying appearance drift through changes in visual attributes such as color and texture. To mitigate these challenges, the paper proposes a functional regularization technique that enables models to incorporate new concepts while preserving prior knowledge. Experimental results validate the effectiveness of this method in reducing semantic and appearance drift, enhancing the stability and reliability of customized models.

Strengths and Weaknesses:

The reviewers generally acknowledge the significance of introducing open-world forgetting in diffusion model customization, along with establishing evaluation metrics and proposing a simple yet effective mitigation strategy, highlighting its contribution to addressing a critical gap in understanding the broader impacts of fine-tuning on model knowledge. However, they express concerns regarding the study's limited contributions and the lack of a convincing evaluation.

In particular, concerns are raised about the study’s applicability to broader datasets, different types of generative models, and more diverse or noisy datasets. The analysis is limited to a small subset of 10 concepts, raising questions about the comprehensiveness of the findings. The evaluation does not incorporate more recent approaches, leaving uncertainties about the generalizability of the method across diverse concepts and models. Additionally, while the paper focuses on how fine-tuning degrades original model capabilities, it does not adequately evaluate the proposed mitigation strategy's impact on fine-tuning task performance. Furthermore, the alignment between quantitative drift assessments and qualitative human evaluations also remain unclear. The paper does not explore whether the identified drifts are exclusive to open-world forgetting or if they also occur in closed-world scenarios.

During the discussion phase, the authors presented a scaled evaluation with a larger set of concepts, demonstrated the applicability of the proposed method to newer approaches with similar improvements, and supplemented their findings with a user study and additional visualizations. However, due to time constraints, the authors were unable to provide complete evaluation results across all datasets. The reviewers remained unconvinced, citing the perceived limited contributions of this work (Reviewer 698f). Additionally, Reviewer 5xU2 raised concerns about the reproducibility of the study, noting discrepancies in DreamBooth's results compared to typical behavior and inferior visualization results in the revised version (Figures 8 and 9) compared to the original submission.

The reviewers and ACs have discussed the paper. Three out of four reviewers suggested boardline acceptance, albeit with some remaining concerns. On the contrary, Reviewer 5xU2 expressed strong reservations. ACs, considering the compelling arguments from Reviewer Yt1v and remaining concerns from other reviewers (Reviewer 698f), lean towards the view that the authors need to conduct more comprehensive evaluations. It is recommended that the authors perform thorough and large-scale evaluations, especially using advanced diffusion models. Additionally, providing a clearer analysis of the results and enhancing missing details would strengthen the paper. Therefore, the paper is not ready for this ICLR. ACs encourage the authors to continue this line of work for future submission.

**Additional Comments On Reviewer Discussion:**

The current recommendation is based on the weaknesses summarized above, particularly the lack of a systematic large-scale evaluation with advanced models to demonstrate the substantial benefits, significance, and reproducibility of the work.

---

### Decision · Program_Chairs · 2025-01-22

Reject